# Certifiably Robust Classifiers: Bridging the Gap Between Theory and Practice

## Abstract

Deep learning models are vulnerable to adversarial attacks, raising important concerns for their use in safety-critical applications. Existing defense methods such as empirical defenses are effective in practice but lack theoretical guarantees, while provable defenses provide a certified robustness radius which is significantly smaller than that achieved by empirical defenses. In this work, we design robust classifiers that leverage the structure of the underlying data distribution, bridging the gap between theoretical certification and strong practical performance. First, we focus on a simple setting where the data distribution is a Gaussian mixture and provide necessary and sufficient conditions under which a robust classifier is guaranteed to exist. We also propose a provably robust classifier along with its certificate of robustness and a generalization guarantee for the learnt certified radius. Next, we generalize our approach to any complex data distribution by using an encoder network to map the input data to a mixture of Gaussians. We also provide a robust classifier with a guaranteed certificate of robustness. Experiments on benchmark datasets indicate that our method outperforms existing top baselines for certified accuracy on CIFAR-10 dataset, while achieving competitive performance on ImageNet even against computationally demanding prior methods.

## 1 Introduction

Deep learning models and AI systems have been shown to be susceptible to adversarial attacks, wherein subtle perturbations to the input data lead to erroneous model outputs (Szegedy et al., 2014). This phenomenon raises serious concerns regarding the deployment of AI models in safety-critical applications. From autonomous vehicles to medical diagnostics and financial decision-making applications, the reliable and trustworthy behavior of the underlying AI models is of enormous importance.

There is an extensive literature on defense methods aiming to defend empirically AI models against existing attacks (Metzen et al., 2017; Papernot et al., 2016; Gu & Rigazio, 2014; Madry et al., 2017; Wong et al., 2020). The currently established methods aim to robustify the underlying model by performing additional training, input preprocessing, denoising filters or even applying multiple defense strategies (Madry et al., 2017; Shafahi et al., 2019; Tramèr et al., 2020). In parallel, the adversarial landscape continues to evolve, with new and more sophisticated attack strategies consistently developed to bypass existing defenses (Athalye et al., 2018; Carlini et al., 2019), creating a recurring cycle between old defenses and new attack strategies.

On the other hand, certified methods have been developed in order to provide provable guarantees of robustness for the underlying models. More specifically, randomized smoothing and its variants (Cohen et al., 2019; Yang et al., 2020; Salman et al., 2022; Chiang et al., 2020) have been the predominant approach offering probabilistic certificates of robustness by smoothing the classifier with Gaussian noise. Beyond randomized smoothing, other certified approaches employ convex relaxations of the adversarial problem (Wong & Kolter, 2018; Raghunathan et al., 2018a), duality-based bounds (Dvijotham et al., 2018), interval bound propagation (Gowal et al., 2018; Zhang et al., 2019), as well as semidefinite programming approaches (Raghunathan et al., 2018b).

While the aforementioned certified defenses offer rigorous robustness guarantees, their development has also created a growing interest in understanding the fundamental principles and limitations be-

Figure 1: The proposed certifiably robust classifier. **Right**: We prove that Gaussian mixture distributions satisfy certain localization properties under which robust classifiers are guaranteed to *exist* (§2), and construct *computable* and *provably robust* classifiers supported on these distributions (§3). **Left**: We extend the guarantees and classifiers to real-world data distributions by leveraging a (locally) Lipschitz encoder that maps real-world distributions to a Gaussian mixture (§4).

hind robust classification. Recent theoretical results identify structural and distributional conditions under which robustness can be provably achieved (Dohmatob, 2019; Shafahi et al., 2018). These results suggest that, in the absence of additional assumptions on the data distribution, certifying robustness may be inherently challenging, motivating a shift toward frameworks that explicitly incorporate distributional knowledge into the certification process. To this end, Pal et al. (2023; 2024) proposed taking into account the intrinsic properties of the data distribution to circumvent existing impossibility results and effectively certify the robustness of deep learning models. In particular, they provide necessary and sufficient conditions under which a robust classifier is guaranteed to exist and a method for constructing certifiably robust classifiers based on the properties of the underlying distribution. Nevertheless, it remains unclear how to verify these conditions in practice for real-world data distributions. Moreover, the proposed classifier is too computationally intensive to be used in large datasets commonly encountered in modern applications.

In this work, we aim to bridge this gap between efficient empirical approaches and existing theoretical methods of certifiable defenses by addressing the following central question:

> *How can we exploit the structure of the data distribution to design classifiers*
> *with provable robustness and strong empirical performance?*

**Paper Contributions.** In this paper, we make the following contributions:

1. *Theoretical Conditions for Robustness in Gaussian Mixtures:* We establish sufficient and practically verifiable theoretical conditions under which a robust classifier is guaranteed to exist for Gaussian mixture distributions. The established results significantly extend the seminal work of Pal et al. (2024), explicitly highlighting the critical role played by geometric properties of the data distribution (e.g., Gaussian mixture means and covariances) in ensuring robustness.

2. *Construction of a Certifiably Robust Classifier with Provable Guarantees:* Building upon our theoretical insights, we construct a robust classifier that leverages the geometry of the underlying data distribution in order to be provably robust against $\ell_2-$norm bounded perturbations. Interestlingly, the constructed classifier is as simple as the well-known quadratic discriminant classification rule. We provide rigorous robustness certificates for the proposed classifier as well as the corresponding generalization bounds, fully characterizing the certified robustness of the proposed classifier.

3. *Generalizing to Complex, Real-world Data Distributions:* We extend our theoretical framework beyond Gaussian mixtures by introducing an encoder network that effectively maps real-world input distributions to a Gaussian mixture distribution. In this way, we are able to utilize our previously derived robust classifier and rigorously establish a theorem for the certified robustness of the whole pipeline for any complex input distribution.

4. *Experimental Evaluation on Benchmark Datasets:* We empirically validate our approach on synthetic and real-world datasets, achieving superior certified accuracy and outperforming the existing state-of-the-art pipelines in certified robustness.

## 2 SUFFICIENT CONDITIONS FOR THE EXISTENCE OF A ROBUST CLASSIFIER FOR GMMS

Consider the setting where the data distribution $\mathcal{D}$ is a Gaussian Mixture Model (GMM) with $K$ components corresponding to the $K$ classes: $\mathcal{D}_i = \mathcal{N}(\mu_i, \Sigma_i)$, $\forall i \in [K]$. We examine the necessary conditions under which each class conditional in the given mixture of Gaussians is $(C, \epsilon, \delta)-$localized with respect to the $\ell_2$ distance. Ensuring that each Gaussian marginal $\mathcal{D}_i$ is localized will satisfy the requirement according to Pal et al. (2023) in showing the existence of a robust classifier.

**Theorem 2.1.** Assume that the data distribution $\mathcal{D}$ is a $d$-dimensional GMM and $\mathcal{D}_i = \mathcal{N}(\mu_i, \Sigma_i)$ corresponds to the class conditional of the $i-$th class. Let $S_i, \forall i \in [K]$, be the ellipsoid set

$$S_i = \{(x - \mu_i)^T \Sigma_i^{-1} (x - \mu_i) \le r_i^2\}. \tag{1}$$

Then, each Gaussian marginal $\mathcal{D}_i$ is $(C, \epsilon, \delta)$-localized on the corresponding set $S_i, \forall i \in [K]$, if and only if the parameters $\epsilon, \delta$ satisfy

$$\delta \le 1 - F_{\chi_d^2}(r_i^2), \quad \epsilon \le \ln\left(\frac{\Gamma\left(\frac{d}{2} + 1\right) C}{\pi^{d/2} r_i^d \sqrt{\det(\Sigma_i)}}\right), \tag{2}$$

where $F_{\chi_d^2}(\cdot)$ is the CDF of the $\chi_d^2-$distribution and $\Gamma(\cdot)$ is the Gamma function.

Theorem 2.1 provides the necessary conditions for the class conditionals $\mathcal{D}_i, \forall i \in [K]$, to be $(C, \epsilon, \delta)$-localized. More specifically, the established conditions are provided for each localization parameter that if satisfied ensure that the Gaussian marginals localize over the aforementioned sets. Importantly, the localization sets resemble the intuition that most points in a Gaussian marginal concentrate around the mean and the shape of the localization set is dictated by the shape of the corresponding covariance matrix $\Sigma_i, \forall i \in [K]$.

The following theorem provides sufficient conditions under which the data distribution $\mathcal{D}$ is $(C, \epsilon, \delta, \gamma)$-strongly localized, which consists of a stronger notion of localization.

**Theorem 2.2.** The data distribution $\mathcal{D}$ is $(C, \epsilon, \delta, \gamma)$-strongly localized with respect to the $\ell_2$ distance, if each class conditional $\mathcal{D}_i = \mathcal{N}(\mu_i, \Sigma_i)$ is $(C, \epsilon, \delta)$-localized on an ellipsoid set $S_i = \{(x - \mu_i)^T \Sigma_i^{-1} (x - \mu_i) \le r_i^2\}$ with parameters

$$\delta \le 1 - F_{\chi_d^2(0)}(r_i^2), \quad \epsilon \le \ln\left(\frac{\Gamma\left(\frac{d}{2} + 1\right) C}{\pi^{d/2} r_i^d \sqrt{\det(\Sigma_i)}}\right), \quad \sum_{j \ne i} F_{\chi_d^2(w_{ij})}(R_j^2) \le \gamma, \tag{3}$$

where $F_{\chi_d^2(w)}$ is the CDF of the $\chi_d^2(w)$ distribution with $d$ degrees of freedom and centrality parameter $w$, $\Gamma$ is the Gamma function, $\lambda_{\min}(\Sigma)$ is the smallest eigenvalue of $\Sigma$,

$$w_{ij} = \|\Sigma_j^{-1/2}(\mu_i - \mu_j)\|_2^2 \quad \text{and} \quad R_j = \frac{2\epsilon}{\sqrt{\lambda_{\min}(\Sigma_j)}} + r_j. \tag{4}$$

Let us pause to elaborate on the implications of the theoretical results established so far. Based on the recent work of Pal et al. (2024), if the data distribution is $(C, \epsilon, \delta, \gamma)$-strongly localized, then there exists an $(\epsilon, \delta)$-robust classifier. However, given that the aforementioned result holds for any distribution $\mathcal{D}$, the previous work of Pal et al. (2024) does not characterize the localization sets nor provides closed-form expressions for each localization parameter. Instead, by focusing on a structured setting instead we are able to define the localization sets $S_i$ and provide in Theorem 2.2 practical sufficient conditions for the existence of a provably robust classifier. On the other hand, (Pal et al., 2024) proves that when classes are balanced, a robust classifier exists only if all class-

conditionals are localized. Thus, using Theorem 2.1 we can provide a way of testing whether there is an $(\epsilon, \delta)$-robust classifier for the underlying data distribution.

# 3 A PROVABLY ROBUST CLASSIFIER FOR $\ell_2$ ATTACKS

In this section, we show how to construct a provably robust classifier against $\ell_2$ attacks for a Gaussian mixture model utilizing the intuition developed in the previous theoretical results. According to the established results of Section 2, if the underlying Gaussian mixture is strongly localized over the ellipsoid sets $S_i$, then a robust classifier is guaranteed to exist. Intuitively the robust classifier should depend on and leverage the structure of the localization sets in order to classify the inputs correctly.

The proposed *nearest ellipsoid* (ELLIPS) classifier operates on ellipsoids $\mathcal{E} = \{E_1, E_2, \ldots, E_K\}$, each one having an associated label $y_i \in \{1, 2, \ldots, K\}$ corresponding to the class $i$, and can be seen as an instantiation of the Bayes classifier for that setting. The ellipsoid $E_i$ is defined by the tuple $(\mu_i, \Sigma_i)_{i \in [K]}$, where $\mu_i, \Sigma_i$ denote the center and covariance matrix of the given ellipsoid. We, also, let $\Pi = (\pi_i)_{i \in [K]}$ be the set of priors of the marginal Gaussian distributions. Having access to the sets $\mathcal{E}, \Pi$, the proposed classifier is given by

$$\text{ELLIPS}(x, \mathcal{E}, \Pi) \quad = \quad y_{i^\star}, \tag{5}$$

where

$$i^\star \quad = \quad \arg\max_{i \in [K]} \{\text{score}(x, E_i, \pi_i)\},$$

$$\text{score}(x, E_i, \pi_i) \quad = \quad -d_M^2(x, E_i) - \log\left(\det(\Sigma_i)\right) + 2\log(\pi_i).$$

and $d_M(x, E_i)$ denotes the Mahalanobis distance of the input sample $x$ from the $i$-th ellipsoid.

The classifier ELLIPS is a nearest ellipsoid classifier with respect to the $d_M$ distance that takes into account two additional terms regarding the shape of the ellipsoid defined by $\Sigma_i$ and the prior $\pi_i$. In this way, the proposed classifier can leverage the geometry of the underlying distribution in order to effectively classify the corresponding input $x \in \mathcal{X}$. Notably, the ELLIPS classifier coincides with the well-known in the literature Quadratic Discriminant Analysis (QDA), whose properties in classification are widely analyzed.

## 3.1 CERTIFICATE OF ROBUSTNESS

In this section, we provide a certificate of robustness against $\ell_2$ adversarial attacks for the ELLIPS classifier. In order to establish theoretical guarantees for the certificate, we first need to define the notion of margin. Formally, the margin of the ELLIPS classifier at a point $x \in \mathcal{X}$ is defined as:

$$m(x) = \text{score}(x, E_{i_*}, \pi_{i_*}) - \text{score}(x, E_{i_2}, \pi_{i_2}),$$

where $i_* = \arg\max_{i \in [K]} \{\text{score}(x, E_i, \pi_i)\}$ and $i_2 = \arg\max_{i \neq i_*} \{\text{score}(x, E_i, \pi_i)\}$ are the classes with the highest and second highest score, respectively. The following theorem provides a certificate of robustness the ELLIPS classifier based on the margin of each point $x \in \mathcal{X}$.

**Theorem 3.1** (Robustness Certificate). Let

$$i_* = \arg\max_{i \in [K]} \{\text{score}(x, E_i, \pi_i)\}, i_2 = \arg\max_{i \neq i_*} \{\text{score}(x, E_i, \pi_i)\}, i_2' = \arg\max_{i \neq i_*} \{\text{score}(x', E_i, \pi_i)\}$$

be the indices of the ellipsoids with the highest and second highest scores for the points $x, x' \in \mathcal{X}$ respectively. Then, we have that

$$\text{ELLIPS}(x, \mathcal{E}, \Pi) = \text{ELLIPS}(x', \mathcal{E}, \Pi)$$

whenever

$$\|x' - x\|_2 \quad \leq \quad \frac{m(x)}{\sqrt{c_M^2 + (-\lambda_{\min})_+ m(x)} + c_M} \tag{6}$$

where $\lambda_{\min}$ is the minimum among all eigenvalues of the matrices $W_i = \Sigma_i^{-1} - \Sigma_{i_*}^{-1}, \forall i \neq i_*$, $(-\lambda_{\min})_+ = \max(-\lambda_{\min}, 0)$, and $c_M = \max_{i \neq i_*} \|\Sigma_{i_*}^{-1}(x - \mu_{i_*})^T - \Sigma_i^{-1}(x - \mu_i)^T\|_2$.

Theorem 3.1 provides a certificate of robustness for the classifier ELLIPS. More specifically, it establishes that the proposed classifier remains robust for any perturbation of size $\epsilon = \|x - x'\|_2$ that is upper bounded by the expression (certified radius) of Theorem 3.1. Importantly, the established certificate of robustness utilizes the geometry around the sample $x$ to be certified and allows for tighter certification based on the local curvature illustrated by $\lambda_{\min}$. Specifically, if $\lambda_{\min} < 0$, Theorem 3.1 provides a second-order certificate, while if $\lambda_{\min} \geq 0$ it resembles a first-order formula for certified radius as common in the literature. Intuitively, the maximum allowed perturbation depends on the margin $m(x)$ of the current point as well as the geometry of the classifier's landscape as indicated by the minimum eigenvalue $\lambda_{\min}$ of the difference of covariance matrices and the term $c_M$.

We note that the form of the certified radius in Theorem 3.1 is a natural consequence of the quadratic nature of the proposed classifier. The proof technique can be utilized to show a similar closed-form for the certificate of robustness of other classical model classes, such as in Linear Discriminant Analysis (LDA) and Support Vector Machines (SVMs) with simple kernels.

### 3.2 ROBUST GENERALIZATION BOUND

In this section, we consider the practical implementation of the ELLIPS classifier and provide a generalization bound for the certificate of robustness of the learnt classifier. So far, we have assumed that the parameters $(\mu_i, \Sigma_i, \pi_i)_{i \in [K]}$ of the ellipsoids are known to analyze the robustness of the proposed classifier. Hereinafter, we consider the learnt classifier ELLIPS which uses the sample mean, sample covariance and class proportions for estimating the true parameters of the underlying distribution. Specifically, if $\{x_j\}_{j=1}^{n_i} \sim \mathcal{D}_i$ are $n_i$ samples from the class $i \in [K]$, the algorithm uses the following estimates

$$\hat{\mu}_i = \frac{1}{n_i} \sum_{j=1}^{n_i} x_j, \quad \hat{\Sigma}_i = \frac{1}{n_i} \sum_{j=1}^{n_i} (x_j - \hat{\mu}_i)(x_j - \hat{\mu}_i)^T, \quad \hat{\pi}_i = \frac{n_i}{n}, \tag{7}$$

where $n = \sum_{i=1}^K n_i$ is the total number of samples. Based on the above estimates, we derive generalization bounds that establish with high probability the robustness of the learnt classifier. More specifically, the following theorem indicates that with high probability the learnt certificate of robustness is close to the true certificate of robustness, thus ensuring the robustness of the learnt classifier ELLIPS.

**Theorem 3.2.** For a sample $(x, y)$, let $\mathcal{R}(x), \hat{\mathcal{R}}(x)$ denote the true and learnt radius of robustness respectively. If the number of samples observed from each Gaussian distribution $\mathcal{D}_i$ is $n = \mathcal{O}\left(\frac{d^{9/2} \log\left(\frac{1}{\delta}\right)}{\epsilon^{9/2}}\right)$, then for any $0 < \epsilon < \epsilon_{\min}$ it holds with probability at least $1 - \delta$ that

$$|\hat{\mathcal{R}}(x) - \mathcal{R}(x)| \leq \mathcal{O}(\epsilon)$$

where $\epsilon_{\min} = \min\{\lambda_{\min}, \lambda_{min}(\Sigma_j), c_M\}$, $\lambda_{\min}$ is the minimum over all eigenvalues of the matrices $W_i = \Sigma_j^{-1} - \Sigma_{j_*}^{-1}, \forall j \neq j_*$ and $\lambda_{\min}(\Sigma_j)$, denotes the minimum eigenvalue value of the covariance matrices $\Sigma_j, \forall j \in [K]$.

Theorem 3.2 provides a generalization bound for the certificate of robustness of the ELLIPS classifier. More specifically, it establishes the required number of samples such that the learnt certified radius of robustness is $\epsilon$-close to the true certified radius of robustness. Interestingly, the provided bound accommondates the change in the expression of (8) based on the local geometry induced at into account both of the closed-form expressions from Theorem 3.1, thus fully characterizing the generalization of the combined formula of certified radius.

## 4 GENERALIZING TO COMPLEX REAL-DATA DISTRIBUTIONS

In this section, we focus on data distributions that are present in real-world applications and might be more complex than the mixture of Gaussians considered thus far. This inherent complexity in the data makes the theoretical guarantees about robustness of an underlying pipeline notoriously to establish in practical situations by explicitly analyzing the underlying distribution.

In this work, we follow a different approach by performing a fine-grained analysis of the properties of a certifiable classifier in the structured and controlled setting of a Gaussian mixture distribution and then generalize our results to construct a provably robust classifier for any encountered data distribution. In particular, we show that by leveraging a neural network $f$ that maps the initial complex data distribution $\mathcal{D}_x$ into a mixture of Gaussians $\mathcal{D}_z$ and then utilizing the already established machinery for the latent Gaussian mixture, a certifiably robust classifier (GENELLIPS) can be instantiated in practice and demonstrably attain competitive certified robust accuracy empirically in benchmark datasets.

We, first, provide a certificate of robustness for the proposed generalized classifier (GENELLIPS) that acts on any arbitrary input distribution and uses an encoder network $f$ that is Lipschitz continuous.

**Theorem 4.1.** Let $f$ be an Lipschitz continuous encoder mapping the input distribution $\mathcal{D}_x$ to a Gaussian mixture $\mathcal{D}_z$. Denote with

$$i_* = \arg\max_{i \in [K]} \{\text{score}(f(x), E_i, \pi_i)\}$$

$$i_2 = \arg\max_{i \neq i_*} \{\text{score}(f(x), E_i, \pi_i)\}, \quad i_2' = \arg\max_{i \neq i_*} \{\text{score}(f(x'), E_i, \pi_i)\}$$

be the indices of the ellipsoids with the highest and second highest scores for two points $x, x' \in \mathcal{X}$ respectively. Then, we have that $\text{ELLIPS}(f(x), \mathcal{E}, \Pi) = \text{ELLIPS}(f(x'), \mathcal{E}, \Pi)$ as long as

$$\|x' - x\|_2 \leq \frac{m(f(x))}{L\left(\sqrt{c_M^2 + (-\lambda_{\min})_+ m(f(x))} + c_M\right)}, \tag{8}$$

where $\lambda_{\min}$ is the minimum among all eigenvalues of the matrices $W_i = \Sigma_i^{-1} - \Sigma_{i_*}^{-1}, \forall i \neq i_*$, $(-\lambda_{\min})_+ = \max(-\lambda_{\min}, 0)$, and $c_M = \max_{i \neq i_*} \|\Sigma_{i_*}^{-1}(f(x) - \mu_{i_*})^T - \Sigma_i^{-1}(f(x) - \mu_i)^T\|_2$.

Theorem 4.1 provides the certified radius of the generalized classifier (GENELLIPS) that uses first the encoder $f$ to map the input data into a Gaussian mixture and then classifies the embedded samples with the ELLIPS classifier. It is important to note that the necessity of a Lipschitz encoder in the above theorem can be relaxed in practice by using an encoder that is locally Lipschitz around the certified sample $x \in \mathcal{X}$. In this way, estimating empirically the local Lipschitz constant $L(x)$ instead of the global one and leveraging the certificate of robustness established in Theorem 4.1, one can effectively compute the certified radius in practice.

It suffices now to select an appropriate encoder network and utilize a certifiable method to estimate the local Lipschitz constant. We provide all the associated details for the implemented pipeline in the next section.

## 5 EXPERIMENTAL EVALUATION

We conduct experiments on both synthetic data and benchmark datasets validating our theoretical results and evaluating the robustness of our proposed classifier in practice.

### 5.1 SYNTHETIC EXPERIMENTS

**Setup.** We conduct experiments in the Gaussian mixture setting, where the input distribution is comprised of $K$ classes and each class is distributed according to $\mathcal{N}(\mu_i, \Sigma_i), \forall i \in [K]$. We run experiments for multiple setups testing for different number of classes $K = \{2, 3, 5, 10\}$ with different

distances $R = \{2, 4, 6\}$ between them, as well as isotropic and non-isotropic covariances matrices $\Sigma$. The means are generated to lie in a circle with angle $\frac{2\pi}{K}$ and radius $R = \{2, 4, 6\}$ from the center in order to control the intersection between the classes. The covariance matrices $\Sigma_i$ are selected to be either isotropic or anisotropic. In the case of isotropic covariances, $\Sigma_i = I$, while in the case of anisotropic covariance matrices the variance in the principal and second principal direction is $1.5, 0.5$ respectively.

**Empirical Validation of Theoretical Results.** We, first, verify empirically our theoretical results established in Sections 2 and 3. We observe that the proposed closed-form expressions for the computation of the localization parameters $\epsilon, \delta, \gamma$ can be used to instantiate the framework established by Pal et al. (2023), where the proposed classifier has certified accuracy equal to $1 - \delta - \gamma$. Simultaneously, they validate the standard intuition of Pal et al. (2023) that data distributions with more distant and non-intersecting classes should have smaller localization parameter $\gamma$, as well as that the localization sets should include most of the mass of the empirical marginal producing a small localization parameter $\delta$. The above observations validate qualitatively our established theoretical results and provide evidence for the usefulness and practicality of our results in instantiating the framework of Pal et al. (2023).

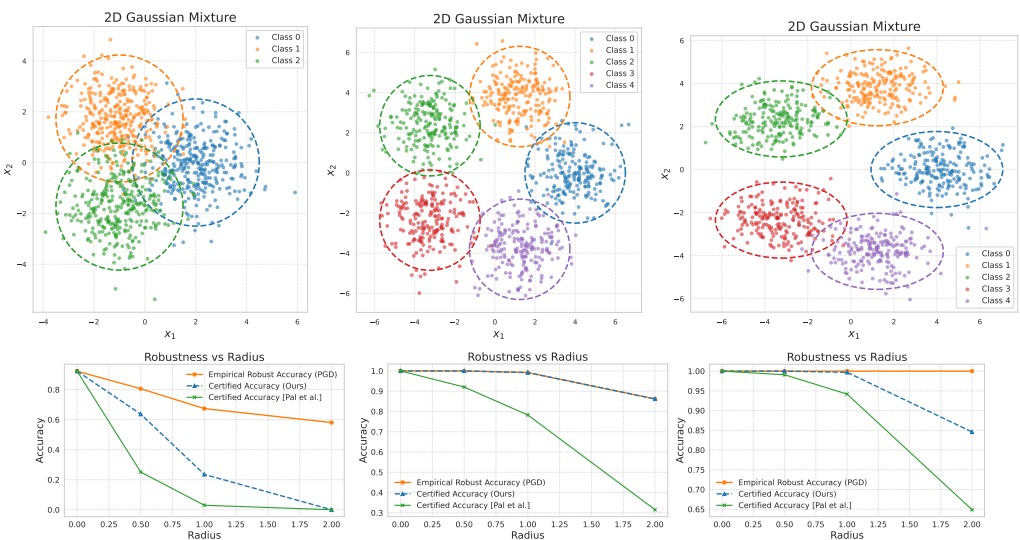

Figure 2: Comparison of different certification methods in Gaussian Mixture distributions. The proposed method outperforms the prior certification scheme of Pal et al. (2023), achieving higher robust accuracy against $\ell_2$ attacks.

**Comparison of Our Method with Pal et al. (2023).** We empirically validate the robustness certificate established for the ELLIPS classifier in Theorem 3.1 and compare the certified accuracy achieved by our method with the framework proposed by Pal et al. (2024). As shown in Figure 2, our approach consistently provides tighter certified robustness guarantees across all experimental settings, significantly outperforming the method of Pal et al. (2023). Furthermore, the certified accuracy for the ELLIPS classifier provided in Theorem 3.1 closely approximates the empirical robust accuracy obtained via PGD attacks, demonstrating the practical tightness of our bound.

**Comparison of Our Method with Randomized Smoothing**. We compare the certified robustness of our method against the widely adopted technique of randomized smoothing. We perform randomized smoothing for different number of samples $n = \{10^3, 10^4, 10^5\}$ and report the certified accuracy achieved in each case. As shown in Figure 3, our method consistently achieves higher certified accuracy across all cases, highlighting the tightness and efficiency of our certification method empirically.

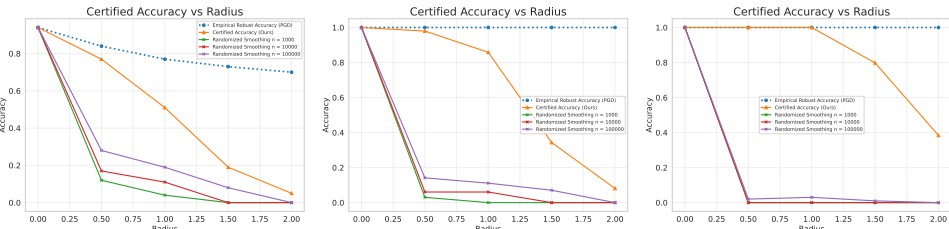

Figure 3: The proposed method outperforms randomized smoothing for different number of samples, producing higher certified accuracy.

## 5.2 EXPERIMENTS ON BENCHMARK DATASETS

**Setup.** We evaluate the robust accuracy of the proposed GENELLIPS classifier on datasets with real-world images. Recall that Theorem 4.1 guarantees the robustness of the proposed classifier by leveraging a Lipschitz encoder that maps the input distribution to a mixture of Gaussians. Thus, to build such a classifier we take a FARE-4 encoder (Schlarmann et al., 2024) pre-trained with an adversarial training objective, and finetune it using an objective promoting isotropy and Gaussianity of the latent distribution. For a detailed description of the loss used for each dataset we refer the interested reader to Appendix 13.1. Additionally, we utilize the CLEVER method (Weng et al., 2018) to estimate the local Lipschitz constant at each sample. Following common practice, we select a confidence interval of $\alpha = 99.9\%$ for estimating the Lipschitz constant of the encoder $f$ via using $n = 1000$ Monte Carlo samples in the CLEVER method.

**Results.** We compare our method against state-of-the-art certified robustness approaches reported in the SoK benchmark by Li et al. (2023) in the CIFAR-10 and ImageNet dataset. As shown in Table 5.2, GENELLIPS consistently outperforms prior methods in the CIFAR-10 dataset, achieving higher certified accuracy across all perturbation levels $\epsilon = \{0.25, 0.5, 1.0\}$. Notably, the classifier simultaneously maintains superior clean accuracy compared to the reported baselines. The presented results highlight the robustness of the proposed pipeline as well as the practical effectiveness of our certification framework.

| Model | Clean Accuracy | Certified Accuracy (%) | | |
|---|---|---|---|---|
| | | $\epsilon = 0.25$ | $\epsilon = 0.5$ | $\epsilon = 1.0$ |
| SmoothAdv (Salman et al., 2019) | 86.2% | 81.0% | 54.4% | 34.8% |
| DRT + MME (Gaussian) (Yang et al., 2022) | 81.4% | 70.4% | 57.8% | 34.4% |
| DRT + MME (SmoothAdv) (Yang et al., 2022) | 72.6% | 67.2% | 60.2% | 39.4% |
| DRT + WE (SmoothAdv) (Yang et al., 2022) | 72.6% | 67.0% | 60.2% | 39.5% |
| **GENELLIPS (Ours)** | **90.14%** | **84.5%** | **78.2%** | **40.5%** |

Table 1: Certified accuracy on CIFAR-10 dataset. The proposed method outperforms the state-of-the-art models in the SoK benchmark, achieving higher robust accuracy without compromising the clean accuracy.

On the ImageNet dataset our method performs competitively against the top baselines for certified accuracy reported in the SoK benchmark. In Table 5.2, the proposed method outperforms all prior baselines with the only exception the ones that utilize diffusion models and thus incur a significantly high computation cost for certification. Specifically, DensePure (Salman et al., 2020) and (Carlini et al., 2023) necessitate performing (multiple) runs of denoising in the attacked image to generate enough samples and then apply a majority vote classifier for certifying robustness. Instead of the computationally cumbersome diffusion process, our method attains competitive robust accuracy by leveraging a pretrained VIP model and using the closed-form certification formula established in our theoretical analysis.

**Wall-clock Time Comparison.** We compare the wall-clock time of the GENELLIPS method with the time that the top diffusion-based benchmarks on ImageNet require. More specifically, Table 5.2

| Model | $\epsilon = 1.0$ | $\epsilon = 2.0$ |
|---|---|---|
| DensePure (Xiao et al., 2022) | **67.0%** | **42.2%** |
| Denoising with Pre-trained Diffusion Models (Carlini et al., 2023) | 54.3% | 29.5% |
| Randomized Smoothing and Adversarial Training (Salman et al., 2020) | 45.0% | 28.0% |
| Ensemble Models and Variance Reduction (Horváth et al., 2022) | 44.6% | 28.6% |
| Ensemble Models (Yang et al., 2022) | 44.4% | 30.4% |
| **GENELLIPS [Ours]** | 45.7% | 31.1% |

Table 2: Certified accuracy on ImageNet dataset. Our approach performs competitively with the models in the SoK benchmark. Demonstrably, it outperforms all state-of-the-art models apart from the ones that use diffusion models, which might be computationally expensive in practice.

indicates that our method is a light-weight approach to certified robustness, trading-off wall-clock time while maintaining competitive performance.

| Model | Wall-clock Time per Image (min) |
|---|---|
| DensePure (Xiao et al., 2022) | 36.47 |
| Denoising with Pre-trained Diffusion Models (Carlini et al., 2023) | 25.599 |
| Randomized Smoothing | 3.015 |
| **GENELLIPS [Ours]** | 2.729 |

Table 3: Wall-clock Time on ImageNet dataset. Our approach requires smaller wall-clock time than the diffusion-based pipelines, offering a computationally light alternative with competitive certified accuracy.

## 6  CONCLUSION

We have proposed a principled framework for leveraging the structure of the data distribution to design classifiers that are both certifiably robust and achieve strong empirical performance. Our theoretical contributions extend prior localization results by providing practical and verifiable conditions for computing localization parameters in Gaussian mixture models, thus ensuring the existence of a robust classifier. Building on the aforementioned results, we introduced a robust classifier that exploits the geometric structure of the underlying distribution and is provably robust against $\ell_2$-adversarial attacks. To handle complex real-world distributions, we generalized our approach using an encoder network that maps inputs to a structured Gaussian mixture, and established a certifiably robust pipeline for any underlying data distribution. Empirical evaluations demonstrated that our method outperforms state-of-the-art robust pipelines, achieving high certified robustness and simultaneously maintaining strong clean accuracy.

There are several promising directions to be considered for future research. One natural extension is to investigate how our framework can be adapted to other adversarial threat models, providing certified classifiers under different $\ell_p$-attacks. Another interesting direction involves developing training strategies that produce encoder networks with lower local Lipschitz constants, thus improving the certified radius under our theoretical guarantees. Overall, we hope our work motivates a new paradigm in certified robustness, whereby the proposed classifiers leverage by design the geometric properties of the data distribution to achieve tighter certified guarantees and simultaneously improved empirical performance.

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

SUPPLEMENTAL MATERIAL

CONTENTS

## 7 ADDITIONAL RELATED WORK ON CERTIFIED ROBUSTNESS

There has been a great line of work on methods establishing theoretical guarantees in the field of certified robustness. The closely related ones to our theoretical investigation aim to correlate the properties of the underlying data distribution with the existence of a robust classifier. In Dohmatob (2019); Pydi & Jog (2020), the authors focus on a binary classification setting and provide a lower bound on the robust classification risk that can be attained. The established bound depends on the Wasserstein distance between the two class conditional distributions, showing that the robust risk increases as the class conditional become closer. This intuition is further extended in the general multi-class classification setting in Pal et al. (2024; 2023) by considering the sets where each marginal distribution localizes and measuring their overlap to estimate the robust risk. However, Pal et al. (2024) do not provide a practical method for computing the associated localization sets, thus constraining the applicability of the established method in practice. Our work instead expands the previous results by providing concrete expressions for the localization sets and the associated parameters and proposing a classifier that utilizes the localization sets in order to robustly classify the input points.

Given that the proposed ELLIPS classifier consists an instantiation of the Bayes classifier for the GMM setting, we provide additional theoretical studies on the optimal Bayes classifier for the clean and adversarial classifier on that setting. Recent work of Dobriban et al. (2023) uses robust isoperimetry and establishes the closed form expressions of the Bayes optimal classifier for the adversarial classification task for two or three classes. The general case even though a fundamental question to the best of our knowledge remains open. Lastly, a specific examination of the classifier for $\ell_0$ attacks is provided in Ashtiani et al. (2020) establishing an asymptotically optimal robust classifier for the GMM setting. We leave as future work examining whether our approach, that uses an encoder and then a classifier for the GMM setting, can be combined with the robust classifier of Ashtiani et al. (2020) to establish robust high certified accuracy results against $\ell_0$ adversarial attacks.

## 8 PROOF OF THEOREM 2.1

*Proof.* In order to prove that each $\mathcal{D}_i, \forall i \in [K]$, is $(C, \epsilon, \delta)$ −localized, we need to show that there is a set $S_i \subseteq \mathcal{X}$ such that the following hold

$$p_i(S_i) \geq 1 - \delta \tag{9}$$
$$\text{Vol}(S_i) \leq Ce^{-\epsilon} \tag{10}$$

where $p_i$ is the density function of $\mathcal{D}_i$. We, first, define the set $S_i$ on which each Gaussian distribution $\mathcal{D}_i$ localizes. To do so, consider the probability density function

$$p_i(x) = \frac{1}{\sqrt{(2\pi)^d \det(\Sigma_i)}} e^{-\frac{1}{2}(x-\mu_i)^T \Sigma_i^{-1}(x-\mu_i)}$$

and the $c$ level-set

$$A_c = \{x \in \mathcal{X} : p_i(x) \geq c\} \tag{11}$$

for some fixed $0 < c < p_i(\mu_i)$. We want to select $c$ such that at least $1 - \delta$ of the mass is included in this level set, so that inequality (9) holds. Note that for a fixed $c$ the level-set is an ellipsoid, as it holds that

$$p_i(x) = c$$
$$\iff \ln p_i(x) = \ln c$$
$$\iff (x - \mu_i)^T \Sigma_i^{-1}(x - \mu_i) = -[2\ln(c) + d\ln(2\pi) + \ln(\det(\Sigma_i))]$$

Letting $r_i^2 = -[2\ln(c) + d\ln(2\pi) + \ln(\det(\Sigma_i))]$ for any $0 < c < p_i(\mu_i)$, the level set in (11) can be equivalently written as

$$S_i = \{x \in \mathcal{X} : (x - \mu_i)^T \Sigma_i^{-1}(x - \mu_i) \leq r_i^2\}$$

which is the set of points with Mahalanobis distance $d_M(x, \mu_i) \leq r_i$.

In order for (9) to hold, we want to find the level set $c$ of $\mathcal{A}_c$ or equivalently the radius $r_i$ of the set $S_i$ such that at least $1 - \delta$ of the mass of the Gaussian distribution $\mathcal{N}(\mu_i, \Sigma_i)$ is included in $S_i$

$$\int_{S_i} p_i(x)dx \geq 1 - \delta \tag{12}$$

The integral in (12) is the probability that a sample $x \sim \mathcal{D}_i$ lies inside the set $S_i$ and thus we get equivalently that the following should hold

$$\mathbb{P}_{x \sim \mathcal{D}_i}(x \in S_i) = \int_{S_i} p_i(x)dx \geq 1 - \delta \tag{13}$$

By a change of variables $y = \Sigma_i^{-1/2}(x - \mu_i)$, we can transform the density $p_i(x)$ inside the integral to the density of the standard $\mathcal{N}(0, I)$ Gaussian $f(x) = \frac{1}{\sqrt{(2\pi)^d}}e^{-\frac{1}{2}\|y\|_2^2}$ and thus the set $S_i$ can be equivalently written as

$$\hat{S}_i = \left\{ x \in \mathcal{X} : \|y\|_2^2 \leq r_i^2 \right\}.$$

Hence, inequality (13) after the change of variables $y = \Sigma_i^{-1/2}(x - \mu_i)$ requires

$$\mathbb{P}_{x \sim \mathcal{D}_i}\left( x \in \mathcal{X} : \|y\|_2^2 \leq r_i^2 \right) \leq 1 - \delta. \tag{14}$$

Note, now, that since $y = \Sigma_i^{-1/2}(x - \mu_i)$ follows the standard Gaussian distribution $\mathcal{N}(0, I)$, the random variable $\|y\|_2^2$ follows the chi-squared distribution with $d$ degrees of freedom. Hence, the left hand-side of (14) is exactly the cumulative probability distribution of the $\chi_d^2$ distribution up to $r_i^2$. Thus, in order for (14) to hold, the $r_i^2$ should be the $(1 - \delta)$-quantile of $\chi_d^2$, i.e.

$$F_{\chi_d^2}(r_i^2) \leq 1 - \delta$$
$$\delta \leq 1 - F_{\chi_d^2}(r_i^2)$$

where $F_{\chi_d^2}$ is the cumulative distribution function of the $\chi_d^2$.

In order for inequality (10) to hold, we have that

$$\text{Vol}(S_i) \leq Ce^{-\epsilon}$$
$$\iff \frac{\pi^{d/2} r_i^d}{\Gamma\left(\frac{d}{2} + 1\right)}\sqrt{\det(\Sigma_i)} \leq Ce^{-\epsilon}$$
$$\iff \epsilon \leq \ln\left( \frac{\Gamma\left(\frac{d}{2} + 1\right) C}{\pi^{d/2} r_i^d \sqrt{\det(\Sigma_i)}} \right) \tag{15}$$

where $\Gamma(\cdot)$ is the Gamma function. $\qquad\square$

## 9  PROOF OF THEOREM 2.2

*Proof.* In order to show that $\mathcal{D}$ is $(C, \epsilon, \delta, \gamma)-$strongly localized, we need to show that for each class conditional $\mathcal{D}_i, \forall i \in [K]$, there is a set $S_i \subseteq \mathcal{X}$ such that the following hold

$$p_i(S_i) \geq 1 - \delta \tag{16}$$
$$\text{Vol}(S_i) \leq Ce^{-\epsilon} \tag{17}$$
$$p_i\left( \bigcup_{j \neq i} S_j^{+2\epsilon} \right) \leq \gamma \tag{18}$$

where the set $S_i^{+\epsilon} = \{x \in \mathcal{X} : \exists \hat{x} \in S_i \text{ with } \|x - \hat{x}\|_2 \leq \epsilon\}$ is the $\epsilon-$expansion of the set $S_i$ with respect to the $\ell_2-$distance. By assumption, we have that the conditions (16), (17) hold. The last condition (inequality (18)) requires that the class conditionals are well-separated in the sense that $p_i\left( \bigcup_{j \neq i} S_j^{+2\epsilon} \right), \forall i \in [K]$, is upper bounded. To ensure that, we can apply the union bound to get

$$p_i\left( \bigcup_{j \neq i} S_j^{+2\epsilon} \right) = \mathbb{P}_{x \sim D_i}\left( \bigcup_{j \neq i} S_j^{+2\epsilon} \right) \leq \sum_{j \neq i} \mathbb{P}_{x \sim D_i}\left( S_j^{+2\epsilon} \right) \tag{19}$$

We, next, bound the probability that a sample $x \sim D_i$ belongs to the set $S_j^{+2\epsilon}$

$$
\begin{aligned}
\mathbb{P}_{x \sim D_i}\left(S_j^{+2\epsilon}\right) &= \mathbb{P}_{x \sim D_i}\left(\exists s \in S_j : \|x - s\|_2 \le 2\epsilon\right) \\
&= \mathbb{P}_{x \sim D_i}\left(\exists s \in \mathcal{X} : d_M(s, \mu_j) \le r_j \text{ and } \|x - s\|_2 \le 2\epsilon\right) \quad (20)
\end{aligned}
$$

Since the expression in (20) involves both the Mahalanobis distance and the $\ell_2$−distance, we will express both conditions $d_M(s, \mu_j) \le r_j$, $\|x - s\|_2 \le 2\epsilon$ in terms of the Mahalanobis distance $d_M(x, \mu_j)$. Using the triangle inequality for the Mahalanobis distance, we get

$$
\begin{aligned}
d_M(x, \mu_j) &\le d_M(x, s) + d_M(s, \mu_j) \\
&\le d_M(x, s) + r_j \\
&= \|\Sigma_j^{-1/2}(x - s)\|_2 + r_j \\
&\le \|\Sigma_j^{-1/2}\|_2 \|x - s\|_2 + r_j \\
&\le \frac{2\epsilon}{\sqrt{\lambda_{\min}(\Sigma_j)}} + r_j \quad (21)
\end{aligned}
$$

where $\lambda_{\min}(\Sigma_j)$ is the smallest eigenvalue of $\Sigma_j$. Hence, for $x \sim D_i$ the event $\mathcal{E}_j = \{\exists s \in \mathcal{X} : d_M(s, D_j) \le r_j \text{ and } \|x - s\|_2 \le 2\epsilon\}$ is contained in the event $\{d_M^2(x, \mu_j) \le R_j^2\}$, where $R_j = \frac{2\epsilon}{\sqrt{\lambda_{\min}(\Sigma_j)}} + r_j$. Thus, we can bound the probability in inequality (20) by

$$
\begin{aligned}
\mathbb{P}_{x \sim D_i}\left(S_j^{+2\epsilon}\right) &\le \mathbb{P}_{x \sim D_i}\left(d_M^2(x, \mu_j) \le R_j^2\right) \\
&= \mathbb{P}_{x \sim D_i}\left((x - \mu_j)^T \Sigma_j^{-1}(x - \mu_j) \le R_j^2\right) \quad (22)
\end{aligned}
$$

Letting $y = \Sigma_j^{-1/2}(x - \mu_j)$, we get from (22) that

$$
\mathbb{P}_{x \sim D_i}\left(S_j^{+2\epsilon}\right) \le \mathbb{P}_{x \sim D_i}\left(\|y\|_2^2 \le R_j^2\right) \quad (23)
$$

Notice that $x \sim \mathcal{D}_i$ and thus the distribution of the random variable $y$ will have mean $\Sigma_j^{-1/2}(\mu_i - \mu_j)$. Thus, the distribution of $\|y\|^2$ will be a non-central $\chi_d^2$−distribution with $d$ degrees of freedom and centrality parameter $w_{ij} = \|\Sigma_j^{-1/2}(\mu_i - \mu_j)\|_2^2$. From inequality (23), we get that

$$
\mathbb{P}_{x \sim D_i}\left(S_j^{+2\epsilon}\right) \le \mathbb{P}_{x \sim D_i}\left(\|y\|_2^2 \le R_j^2\right) = F_{\chi_d^2(w_{ij})}(R_j^2) \quad (24)
$$

where $F_{\chi_d^2(w_{ij})}(\cdot)$ is the cumulative density function of the $\chi_d^2(w_{ij})$ distribution.

Substituting inequality (24) into (19), we obtain the following bound

$$
\begin{aligned}
p_i\left(\bigcup_{j \neq i} S_j^{+2\epsilon}\right) &\le \sum_{j \neq i} \mathbb{P}_{x \sim D_i}\left(S_j^{+2\epsilon}\right) \\
&\le \sum_{j \neq i} F_{\chi_d^2(w_{ij})}(R_j^2)
\end{aligned}
$$

and thus letting $\gamma = \sum_{j \neq i} F_{\chi_d^2(w_{ij})}(R_j^2)$ finishes the proof. □

## 10 PROOF OF THEOREM 3.1

*Proof.* Consider a sample $(x, y)$ with positive margin

$$
m(x) = \text{score}(x, E_y, \pi_y) - \max_{i \neq i_*} \text{score}(x, E_i, \pi_i) > 0 \quad (25)
$$

where $i_* = \max_{i \in [K]} \text{score}(x, E_i, \pi_i) = y$.

We want to show that the perturbed sample $x'$ has also positive margin

$$m(x') = \text{score}(x', E_{i_*}, \pi_{i_*}) - \max_{i \neq i_*} \text{score}(x', E_i, \pi_i) \tag{26}$$

Substituting the definition of score and rearranging the terms, we have

$$
\begin{aligned}
m(x') &= -(x' - \mu_{i_*})\Sigma_{i_*}^{-1}(x' - \mu_{i_*})^T - \log\left(\det(\Sigma_{i_*})\right) + 2\log(\pi_{i_*}) \\
&\quad - \max_{i \neq i_*}\{-(x' - \mu_i)\Sigma_i^{-1}(x' - \mu_i)^T - \log\left(\det(\Sigma_i)\right) + 2\log(\pi_i)\} \\
&= \min_{i \neq i_*}\Big\{ -(x' - \mu_{i_*})\Sigma_{i_*}^{-1}(x' - \mu_{i_*})^T + (x' - \mu_i)\Sigma_i^{-1}(x' - \mu_i)^T \\
&\qquad\qquad - \log\left(\frac{\det(\Sigma_{i_*})}{\det(\Sigma_i)}\right) + 2\log\left(\frac{\pi_{i_*}}{\pi_i}\right) \Big\}
\end{aligned}
\tag{27}
$$

For any $x, x' \in \mathcal{X}$ and $\forall i \in [K]$, we have that

$$
\begin{aligned}
(x' - \mu_i)\Sigma_i^{-1}(x' - \mu_i)^T &= (x' - x)\Sigma_i^{-1}(x' - \mu_i)^T + (x - \mu_i)\Sigma_i^{-1}(x' - \mu_i)^T \\
&= (x' - x)\Sigma_i^{-1}(x' - x)^T + 2(x' - x)\Sigma_i^{-1}(x - \mu_i)^T \\
&\quad + (x - \mu_i)\Sigma_i^{-1}(x - \mu_i)^T
\end{aligned}
\tag{28}
$$

Using (28) into (27) for the terms $-(x' - \mu_{i_*})\Sigma_{i_*}^{-1}(x' - \mu_{i_*})^T$ and $(x' - \mu_i)\Sigma_i^{-1}(x' - \mu_i)^T$, we have that

$$
\begin{aligned}
m(x') &= \min_{i \neq i_*}\Big\{ -(x' - x)\Sigma_{i_*}^{-1}(x' - x)^T - 2(x' - x)\Sigma_{i_*}^{-1}(x - \mu_{i_*})^T - (x - \mu_{i_*})\Sigma_{i_*}^{-1}(x - \mu_{i_*})^T \\
&\qquad + (x' - x)\Sigma_i^{-1}(x' - x)^T + 2(x' - x)\Sigma_i^{-1}(x - \mu_i)^T + (x - \mu_i)\Sigma_i^{-1}(x - \mu_i)^T \\
&\qquad - \log\left(\frac{\det(\Sigma_{i_*})}{\det(\Sigma_i)}\right) + 2\log\left(\frac{\pi_{i_*}}{\pi_i}\right) \Big\} \\
&= \min_{i \neq i_*}\Big\{ (x' - x)(\Sigma_i^{-1} - \Sigma_{i_*}^{-1})(x' - x)^T + 2(x' - x)[\Sigma_i^{-1}(x - \mu_i)^T - \Sigma_{i_*}^{-1}(x - \mu_{i_*})^T] \\
&\qquad - (x - \mu_{i_*})\Sigma_{i_*}^{-1}(x - \mu_{i_*})^T + (x - \mu_i)\Sigma_i^{-1}(x - \mu_i)^T \\
&\qquad - \log\left(\frac{\det(\Sigma_{i_*})}{\det(\Sigma_i)}\right) + 2\log\left(\frac{\pi_{i_*}}{\pi_i}\right) \Big\}
\end{aligned}
\tag{29}
$$

Given that the matrix $W_i = \Sigma_i^{-1} - \Sigma_{i_*}^{-1}$ is symmetric, as the difference of inverses of symmetric matrices, it holds that $(x' - x)W_i(x' - x)^T \geq \lambda_{\min}(W_i)\|x' - x\|_2^2$. Using that and Cauchy-Schwarz inequality, we get from (29)

$$
\begin{aligned}
m(x') &\geq \min_{i \neq i_*}\Big\{ \lambda_{\min}(W_i)\|x' - x\|_2^2 - 2\|x' - x\|_2\|\Sigma_i^{-1}(x - \mu_i)^T - \Sigma_{i_*}^{-1}(x - \mu_{i_*})^T\|_2 \\
&\qquad - (x - \mu_{i_*})\Sigma_{i_*}^{-1}(x - \mu_{i_*})^T + (x - \mu_i)\Sigma_i^{-1}(x - \mu_i)^T \\
&\qquad + \log\left(\frac{\det(\Sigma_i)}{\det(\Sigma_{i_*})}\right) + 2\log\left(\frac{\pi_{i_*}}{\pi_i}\right) \Big\}
\end{aligned}
\tag{30}
$$

Using the subadditivity of the min operator and the definition of margin $m(x)$ from (25), we get

$$m(x') \geq m(x) + \min_{i \neq i_*} \lambda_{\min}(W_i)\|x' - x\|_2^2 - 2\|x' - x\|_2 \max_{i \neq i_*} \|\Sigma_{i_*}^{-1}(x - \mu_{i_*})^T - \Sigma_i^{-1}(x - \mu_i)^T\|_2$$

Letting for brevity $\lambda_{\min} = \min_{i \neq i_*}\{\lambda_{\min}(W_i)\}$ and $c_M = \max_{i \neq i_*} \|\Sigma_{i_*}^{-1}(x - \mu_{i_*})^T - \Sigma_i^{-1}(x - \mu_i)^T\|_2$, in order for $m(x')$ to be non-negative, it suffices that

$$m(x) + \lambda_{\min}\|x' - x\|_2^2 - 2c_M\|x' - x\|_2 \quad > \quad 0$$

If $\lambda_{\min} < 0$, then we have that

$$\|x' - x\|_2 \quad \leq \quad \frac{c_M - \sqrt{c_M^2 - m(x)\lambda_{\min}}}{\lambda_{\min}} = \frac{m(x)}{c_M + \sqrt{c_M^2 - m(x)\lambda_{\min}}} \tag{31}$$

If $\lambda_{\min} \geq 0$, then it suffices that

$$\|x' - x\|_2 \quad \leq \quad \frac{m(x)}{2c_M}. \tag{32}$$

Combining the expressions in (31) and (32), we get the final result. $\qquad\square$

## 11 PROOF OF GENERALIZATION BOUND

We, first, provide some necessary Lemmas in Section 11.1 bounding the associated quantities appearing in the generalization bound and then we provide the proof of Theorem 3.2 in Section 11.2.

### 11.1 PREPARATORY LEMMAS

**Lemma 11.1.** For a Gaussian marginal with true mean $\mu$ and covariance matrix $\Sigma$ and empirical mean and covariance $\hat{\mu}, \hat{\Sigma}$ satisfying $\|\hat{\mu} - \mu\|_{\Sigma^{-1}} \leq \tilde{\mathcal{O}}\left(\sqrt{\frac{d}{n}}\right), \|\hat{\Sigma} - \Sigma\|_{op} \leq \tilde{\mathcal{O}}\left(\sqrt{\frac{d}{n}}\right)$, we have that for any point $x \in \mathcal{X}$ it holds that

$$|d_M^2(x, \mu) - d_M^2(x, \hat{\mu})| \leq \tilde{\mathcal{O}}\left(\frac{d^{3/2}}{n^{3/2}}\right)$$

where $d_M(x, \mu)$ corresponds to the Mahalanobis distance.

*Proof.* We have that

$$|d_M^2(x, \mu) - d_M^2(x, \hat{\mu})| = \left|(x - \mu)^T \Sigma^{-1}(x - \mu) - (x - \hat{\mu})^T \hat{\Sigma}^{-1}(x - \hat{\mu})\right|$$

Adding and subtracting the term $(x - \hat{\mu})^T \Sigma^{-1}(x - \hat{\mu})$ and applying the triangle inequality, we get

$$|d_M^2(x, \mu) - d_M^2(x, \hat{\mu})| = \left|(x - \mu)^T \Sigma^{-1}(x - \mu) - (x - \hat{\mu})^T \Sigma^{-1}(x - \hat{\mu})\right.$$
$$\left. + (x - \hat{\mu})^T \Sigma^{-1}(x - \hat{\mu}) - (x - \hat{\mu})^T \hat{\Sigma}^{-1}(x - \hat{\mu})\right|$$
$$\leq \underbrace{\left|(x - \mu)^T \Sigma^{-1}(x - \mu) - (x - \hat{\mu})^T \Sigma^{-1}(x - \hat{\mu})\right|}_{T_1}$$
$$+ \underbrace{\left|(x - \hat{\mu})^T \left(\hat{\Sigma}^{-1} - \Sigma^{-1}\right)(x - \hat{\mu})\right|}_{T_2} \tag{33}$$

We, next, bound the two terms $T_1, T_2$. By rearranging the terms in $T_1$, we get

$$T_1 = \left|(x - \mu)^T \Sigma^{-1}(x - \mu) - (x - \hat{\mu})^T \Sigma^{-1}(x - \hat{\mu})\right|$$
$$= \left|(\hat{\mu} - \mu)^T \Sigma^{-1} x - (x - \mu)^T \Sigma^{-1} \mu + (x - \hat{\mu})^T \Sigma^{-1} \hat{\mu}\right|$$
$$= \left|(\hat{\mu} - \mu)^T \Sigma^{-1} x - (x - \mu)^T \Sigma^{-1} \mu + (x - \hat{\mu})^T \Sigma^{-1}(\hat{\mu} - \mu) + (x - \hat{\mu})^T \Sigma^{-1} \mu\right|$$
$$= \left|2\Delta\hat{\mu}^T \Sigma^{-1}(x - \mu) + \Delta\hat{\mu}^T \Sigma^{-1} \Delta\hat{\mu}\right|$$
$$\leq 2\|\Delta\hat{\mu}\|_{\Sigma^{-1}}\|x - \mu\|_{\Sigma^{-1}} + \|\Delta\hat{\mu}\|_{\Sigma^{-1}}^2$$

Using the fact that $\|\Delta\hat{\mu}\|_{\Sigma^{-1}} = \tilde{\mathcal{O}}\left(\sqrt{\frac{d}{n}}\right)$, we obtain

$$T_1 \leq \tilde{\mathcal{O}}\left(\frac{d}{n}\right) \tag{34}$$

We can bound the term $T_2$ using the inequality

$$T_2 = \left|(x - \hat{\mu})^T \left(\hat{\Sigma}^{-1} - \Sigma^{-1}\right)(x - \hat{\mu})\right| \leq \|x - \hat{\mu}\|_2^2 \|\hat{\Sigma}^{-1} - \Sigma^{-1}\|_{op}$$

Adding and subtracting the true mean $\mu$ and utilizing the definition of $\Delta\hat{\mu} = \hat{\mu} - \mu$, we get

$$T_2 \leq \|x - \mu + \Delta\hat{\mu}\|_2^2 \|\hat{\Sigma}^{-1} - \Sigma^{-1}\|_{op}$$
$$\leq 2\left(\|x - \mu\|_2^2 + \|\Delta\hat{\mu}\|_2^2\right)\|\hat{\Sigma}^{-1} - \Sigma^{-1}\|_{op}$$

where at the last step we have applied the inequality $\|a + b\|^2 \leq 2\|a\|^2 + 2\|b\|^2$. Using the fact that $\|\Delta\hat{\mu}\|_2 = \tilde{\mathcal{O}}\left(\sqrt{\frac{d}{n}}\right), \|\hat{\Sigma}^{-1} - \Sigma^{-1}\|_{op} = \tilde{\mathcal{O}}\left(\sqrt{\frac{d}{n}}\right)$, we obtain that

$$T_2 \leq \tilde{\mathcal{O}}\left(\frac{d^{3/2}}{n^{3/2}}\right) \tag{35}$$

Substituting inequalities (34), (35) into (33), we have that

$$|d_M^2(x, \mu) - d_M^2(x, \hat{\mu})| \leq \tilde{\mathcal{O}}\left(\frac{d}{n}\right) + \tilde{\mathcal{O}}\left(\frac{d^{3/2}}{n^{3/2}}\right) \leq \tilde{\mathcal{O}}\left(\frac{d^{3/2}}{n^{3/2}}\right)$$

□

**Lemma 11.2.** For any symmetric positive definite matrices $A, B$ with $\lambda_{\min}(A) > \|B - A\|_{\mathrm{op}}$, it holds that

$$|\log \det(A) - \log \det(B)| \leq \frac{d\|B - A\|_{\mathrm{op}}}{\lambda_{\min}(A) - \|B - A\|_{\mathrm{op}}}$$

*Proof.* From the trace representation, we have that for any two symmetric positive definite matrices $A, B$ it holds that

$$|\log \det(A) - \log \det(B)| = \left|\mathrm{Tr}\left(\int_0^1 (A + t(B - A))^{-1} (B - A) dt\right)\right|$$

$$\leq \frac{1}{\lambda_{\min}(A) - \|B - A\|_{\mathrm{op}}}\|B - A\|_{\mathrm{op}} \qquad (36)$$

Using triangle inequality, we get

$$|\log \det(A) - \log \det(B)| \leq \int_0^1 \left|\mathrm{Tr}\left((A + t(B - A))^{-1} (B - A)\right)\right| dt \qquad (37)$$

From Holder's inequality for trace we have that for any two matrices $X, Y$ it holds that $|\mathrm{Tr}(XY)| \leq d\|X\|_{\mathrm{op}}\|Y\|_{\mathrm{op}}$. Applying that for $X = (A + t(B - A))^{-1}$ and $Y = B - A$, we obtain

$$\left|\mathrm{Tr}\left((A + t(B - A))^{-1} (B - A)\right)\right| \leq d\|B - A\|_{\mathrm{op}}\left\|(A + t(B - A))^{-1}\right\|_{\mathrm{op}} \qquad (38)$$

Substituting (38) into (37), we obtain

$$|\log \det(A) - \log \det(B)| \leq d\|B - A\|_{\mathrm{op}}\int_0^1 \left\|(A + t(B - A))^{-1}\right\|_{\mathrm{op}} dt \qquad (39)$$

We, now, bound the operator norm $\left\|(A + t(B - A))^{-1}\right\|_{\mathrm{op}}$. Using the fact that $\left\|(A + t(B - A))^{-1}\right\|_{\mathrm{op}} \leq \frac{1}{\lambda_{\min}(A + t(B - A))}$ and Weyl's inequality we have that $\lambda_{\min}(A + t(B - A)) \geq \lambda_{\min}(A) + \|B - A\|_{\mathrm{op}}, \forall t \in [0, 1]$. Thus, we get

$$\left\|(A + t(B - A))^{-1}\right\|_{\mathrm{op}} \leq \frac{1}{\lambda_{\min}(A) - \|B - A\|_{\mathrm{op}}} \qquad (40)$$

Substituting (40) into (39), we have

$$|\log \det(A) - \log \det(B)| \leq d\|B - A\|_{\mathrm{op}}\int_0^1 \frac{dt}{\lambda_{\min}(A) - \|B - A\|_{\mathrm{op}}} = \frac{d\|B - A\|_{\mathrm{op}}}{\lambda_{\min}(A) - \|B - A\|_{\mathrm{op}}}$$

□

**Lemma 11.3.** If the number of samples observed from two Gaussian marginals $\mathcal{D}_{i_*}, \mathcal{D}_{i_2'}$ is at least $n \geq d$ and $\|\Sigma_{i_*} - \hat{\Sigma}_{i_*}\|_{\mathrm{op}} < \lambda_{\min}(\Sigma_{i_*}), \|\Sigma_{i_*} - \hat{\Sigma}_{i_*}\|_{\mathrm{op}} < \lambda_{\min}(\Sigma_{i_2'})$, the following holds

$$|\log \det(\Sigma_{i_*}) - \log \det(\hat{\Sigma}_{i_*})| + |\log \det(\Sigma_{i_2'}) - \log \det(\hat{\Sigma}_{i_2'})| \leq \tilde{\mathcal{O}}\left(\frac{d}{n}\right)$$

*Proof.* From Lemma 11.2, for $A = \Sigma_{i_*}, B = \hat{\Sigma}_{i_*}$ and $\epsilon = \|\Sigma_{i_*} - \hat{\Sigma}_{i_*}\|_{\mathrm{op}}$, we get

$$|\log\det(\Sigma_{i_*}) - \log\det(\hat{\Sigma}_{i_*})| \leq \frac{d\epsilon}{\lambda_{\min}(\Sigma_{i_*})(1 - \epsilon/\lambda_{\min}(\Sigma_{i_*}))} \tag{41}$$

Given that $\epsilon = \|\Sigma_{i_*} - A\|_{\mathrm{op}} = \tilde{\mathcal{O}}\left(\sqrt{\frac{d}{n}}\right)$, we get for $\epsilon < \lambda_{\min}(\Sigma_{i_*})$ from the Taylor expansion of the function $f(x) = (1 - \frac{1}{x})^{-1} = \sum_{n=0}^{+\infty} x^n$ that

$$(1 - \epsilon/\lambda_{\min}(\Sigma_{i_*}))^{-1} \leq \sum_{n=0}^{+\infty}\left(\frac{\epsilon}{\lambda_{\min}(\Sigma_{i_*})}\right)^n = \tilde{\mathcal{O}}\left(\epsilon\right) \tag{42}$$

Substituting (42) into (41), we get that

$$|\log\det(\Sigma_{i_*}) - \log\det(\hat{\Sigma}_{i_*})| \leq \tilde{\mathcal{O}}\left(\epsilon^2\right) = \tilde{\mathcal{O}}\left(\frac{d}{n}\right) \tag{43}$$

By applying the above steps similarly for $A = \Sigma_{i_2'}$ and $B = \hat{\Sigma}_{i_2'}$ and using the fact that $\|\Sigma_{i_2'} - \hat{\Sigma}_{i_2'}\|_{\mathrm{op}} = \tilde{\mathcal{O}}\left(\sqrt{\frac{d}{n}}\right)$, we obtain

$$|\log\det(\Sigma_{i_2'}) - \log\det(\hat{\Sigma}_{i_2'})| \leq \tilde{\mathcal{O}}\left(\frac{d}{n}\right) \tag{44}$$

Adding inequalities (43), (44), we get the final result

$$|\log\det(\Sigma_{i_*}) - \log\det(\hat{\Sigma}_{i_*})| + |\log\det(\Sigma_{i_2'}) - \log\det(\hat{\Sigma}_{i_2'})| \leq \tilde{\mathcal{O}}\left(\frac{d}{n}\right)$$

$\square$

**Lemma 11.4.** If the number of samples observed from two Gaussian marginals $\mathcal{D}_{i_*}, \mathcal{D}_{i_2'}$ is at least $n \geq d$ and $|\pi_{i_*} - \hat{\pi}_{i_*}| < 1, |\pi_{i_2'} - \hat{\pi}_{i_2'}| < 1$, then it holds that

$$|\log\left(\hat{\pi}_{i_2'}\right) - \log\left(\pi_{i_2'}\right)| + |\log\left(\hat{\pi}_{i_*}\right) - \log\left(\pi_{i_*}\right)| \leq \tilde{\mathcal{O}}\left(\sqrt{\frac{d}{n}}\right)$$

*Proof.* For a Gaussian distribution with true prior $\pi$ and empirical prior $\hat{\pi}$ from the Taylor expansion we get that for $|\pi - \hat{\pi}| < 1$ it holds

$$\log\left(\frac{\hat{\pi}}{\pi}\right) = \sum_{i=1}^{+\infty} \frac{(-1)^{i-1}}{i}\left(\frac{\hat{\pi}}{\pi} - 1\right)^i$$

Additionally, given that the empirical prior approaches the true prior as $|\pi - \hat{\pi}| \leq \tilde{\mathcal{O}}\left(\sqrt{\frac{d}{n}}\right)$ with $d \leq n$, we have that

$$\log\left(\frac{\hat{\pi}}{\pi}\right) \leq \tilde{\mathcal{O}}\left(\frac{\hat{\pi} - \pi}{\pi}\right) = \tilde{\mathcal{O}}\left(\sqrt{\frac{d}{n}}\right) \tag{45}$$

Applying inequality (45) for the Gaussian marginal $\mathcal{D}_{i_2'}$, we get

$$\log\left(\frac{\hat{\pi}_{i_2'}}{\pi_{i_2'}}\right) \leq \tilde{\mathcal{O}}\left(\sqrt{\frac{d}{n}}\right)$$

$$|\log\left(\hat{\pi}_{i_2'}\right) - \log\left(\pi_{i_2'}\right)| \leq \tilde{\mathcal{O}}\left(\sqrt{\frac{d}{n}}\right) \tag{46}$$

Similarly, applying (45) for $\mathcal{D}_{i_*}$, we have that

$$\log\left(\frac{\hat{\pi}_{i_*}}{\pi_{i_*}}\right) \leq \tilde{\mathcal{O}}\left(\sqrt{\frac{d}{n}}\right)$$

$$|\log\left(\hat{\pi}_{i_*}\right) - \log\left(\pi_{i_*}\right)| \leq \tilde{\mathcal{O}}\left(\sqrt{\frac{d}{n}}\right) \tag{47}$$

Adding inequalities (46), (47), we obtain

$$\left| \log\left(\hat{\pi}_{i_2'}\right) - \log\left(\pi_{i_2'}\right)\right| + \left| \log\left(\hat{\pi}_{i_*}\right) - \log\left(\pi_{i_*}\right)\right| \;\leq\; \tilde{\mathcal{O}}\left(\sqrt{\frac{d}{n}}\right)$$

□

**Lemma 11.5.** Let the functions

$$\mathrm{m}(x) = d_M^2(x, \mu_{i_2'}) - d_M^2(x, \mu_{i_*}) + \log\left(\frac{\det\left(\Sigma_{i_2'}\right)}{\det\left(\Sigma_{i_*}\right)}\right) - 2\log\left(\frac{\pi_{i_2'}}{\pi_{i_*}}\right),$$

$$\hat{\mathrm{m}}(x) = d_M^2(x, \hat{\mu}_{i_2'}) - d_M^2(x, \hat{\mu}_{i_*}) + \log\left(\frac{\det\left(\hat{\Sigma}_{i_2'}\right)}{\det\left(\hat{\Sigma}_{i_*}\right)}\right) - 2\log\left(\frac{\hat{\pi}_{i_2'}}{\hat{\pi}_{i_*}}\right).$$

where $\hat{\mu}_j, \hat{\Sigma}_j$ is the empirical mean and covariance of the Gaussian marginal $\mathcal{D}_j, \forall j \in [K]$. If the number of samples observed from each marginal $\mathcal{D}_j, \forall j \in [K]$ is $n \geq d$, then we have that

$$|\hat{m}(x) - m(x)| \;\leq\; \tilde{\mathcal{O}}\left(\frac{d^{3/2}}{n^{3/2}}\right)$$

*Proof.* Using the expressions of margins and applying the triangle inequality, we have that

$$
\begin{aligned}
|\hat{\mathrm{m}}(x) - \mathrm{m}(x)| &= \left| d_M^2(x, \hat{\mu}_{i_2'}) - d_M^2(x, \mu_{i_2'}) - d_M^2(x, \hat{\mu}_{i_*}) + d_M^2(x, \mu_{i_*}) \right. \\
&\quad \left. + \log\left(\frac{\det\left(\hat{\Sigma}_{i_2'}\right)}{\det\left(\hat{\Sigma}_{i_*}\right)}\right) - \log\left(\frac{\det\left(\Sigma_{i_2'}\right)}{\det\left(\Sigma_{i_*}\right)}\right) - 2\log\left(\frac{\hat{\pi}_{i_2'}}{\hat{\pi}_{i_*}}\right) + 2\log\left(\frac{\pi_{i_2'}}{\pi_{i_*}}\right)\right| \\
&\leq \underbrace{\left| d_M^2(x, \hat{\mu}_{i_2'}) - d_M^2(x, \mu_{i_2'})\right| + \left| d_M^2(x, \hat{\mu}_{i_*}) - d_M^2(x, \mu_{i_*})\right|}_{T_1} \\
&\quad + \underbrace{\left|\log\left(\det\left(\hat{\Sigma}_{i_2'}\right)\right) - \log\left(\det\left(\Sigma_{i_2'}\right)\right)\right| + \left|\log\left(\det\left(\hat{\Sigma}_{i_*}\right)\right) - \log\left(\det\left(\Sigma_{i_*}\right)\right)\right|}_{T_2} \\
&\quad + 2\underbrace{\left[\left|\log\left(\hat{\pi}_{i_2'}\right) - \log\left(\pi_{i_2'}\right)\right| + \left|\log\left(\hat{\pi}_{i_*}\right) - \log\left(\pi_{i_*}\right)\right|\right]}_{T_3} \tag{48}
\end{aligned}
$$

We, next, bound the three terms $T_1, T_2, T_3$. Applying Lemma 11.1 for the Gaussian marginals $\mathcal{D}_{i_2}, \mathcal{D}_{i_*}$, we have that

$$\left| d_M^2(x, \hat{\mu}_{i_2'}) - d_M^2(x, \mu_{i_2'})\right| \;\leq\; \tilde{\mathcal{O}}\left(\frac{d}{n}\right) + \tilde{\mathcal{O}}\left(\frac{d^{3/2}}{n^{3/2}}\right) \tag{49}$$

$$\left| d_M^2(x, \hat{\mu}_{i_*}) - d_M^2(x, \mu_{i_*})\right| \;\leq\; \tilde{\mathcal{O}}\left(\frac{d}{n}\right) + \tilde{\mathcal{O}}\left(\frac{d^{3/2}}{n^{3/2}}\right) \tag{50}$$

Adding inequalities (49), (50), we get that

$$T_1 \;\leq\; \tilde{\mathcal{O}}\left(\frac{d^{3/2}}{n^{3/2}}\right) \tag{51}$$

We, next, bound the term $T_2$ by using Lemma 11.3

$$T_2 \leq \tilde{\mathcal{O}}\left(\frac{d}{n}\right) \tag{52}$$

For the term $T_3$, we use Lemma 11.4 and get

$$T_3 \;\leq\; \tilde{\mathcal{O}}\left(\sqrt{\frac{d}{n}}\right) \tag{53}$$

Substituting inequalities (51), (52), (53) into (48), we obtain

$$|\hat{m}(x) - m(x)| \leq \tilde{\mathcal{O}}\left(\frac{d^{3/2}}{n^{3/2}}\right) + \tilde{\mathcal{O}}\left(\sqrt{\frac{d}{n}}\right) + 2\tilde{\mathcal{O}}\left(\sqrt{\frac{d}{n}}\right) \leq \tilde{\mathcal{O}}\left(\frac{d^{3/2}}{n^{3/2}}\right)$$

$\square$

**Lemma 11.6.** If the number of samples observed from each marginal $\mathcal{D}_j, \forall j \in [K]$, is at least $n \geq d$, then we have that

$$\left|\lambda_{\min} - \hat{\lambda}_{\min}\right| \leq \tilde{\mathcal{O}}\left(\sqrt{\frac{d}{n}}\right)$$

where $\lambda_{\min}$ is the minimum over all eigenvalues of $W_i = \Sigma_i^{-1} - \Sigma_{i_*}^{-1}, \forall i \neq i_*$.

*Proof.* Using the fact that for any two real sequences $a_i, b_i$ it holds that $|\min_i a_i - \min_i b_i| \leq \max_i |a_i - b_i|$, we have that

$$\left|\lambda_{\min} - \hat{\lambda}_{\min}\right| = \left|\min_{i \in [K]} \lambda_{\min}(W_i) - \min_{i \in [K]} \lambda_{\min}(\hat{W}_i)\right|$$

$$\leq \max_{i \in [K]} \left|\lambda_{\min}(W_i) - \lambda_{\min}(\hat{W}_i)\right| \tag{54}$$

$$\tag{55}$$

Using Weyl's and triangle inequality, we get

$$\left|\lambda_{\min} - \hat{\lambda}_{\min}\right| \leq \max_i \left\|W_i - \hat{W}_i\right\|_{\mathrm{op}} \tag{56}$$

$$= \max_i \left\|(\Sigma_i^{-1} - \Sigma_*^{-1}) - (\hat{\Sigma}_i^{-1} - \hat{\Sigma}_*^{-1})\right\|_{\mathrm{op}} \tag{57}$$

$$\leq \max_i \left(\|\Sigma_i^{-1} - \hat{\Sigma}_i^{-1}\|_{\mathrm{op}} + \|\Sigma_*^{-1} - \hat{\Sigma}_*^{-1}\|_{\mathrm{op}}\right) \tag{58}$$

$$\leq \max_i \left(\|\Sigma_i^{-1}\|_{\mathrm{op}}^2 \|\Sigma_i - \hat{\Sigma}_i\|_{\mathrm{op}} + \|\Sigma_*^{-1}\|_{\mathrm{op}}^2 \|\Sigma_* - \hat{\Sigma}_*\|_{\mathrm{op}}\right) \tag{59}$$

$$= \tilde{\mathcal{O}}\left(\sqrt{d/n}\right) \tag{60}$$

where at the last step we used the fact that $\|\Sigma_i - \hat{\Sigma}_i\|_{\mathrm{op}} = \tilde{\mathcal{O}}\left(\sqrt{\frac{d}{n}}\right)$. $\square$

**Lemma 11.7.** If the number of samples observed from each marginal $\mathcal{D}_j, \forall j \in [K]$, is at least $n \geq d$ and $|\hat{\lambda}_{\min} - \lambda_{\min}| < \lambda_{\min}$, then it holds that

$$\left|\frac{1}{\hat{\lambda}_{\min}} - \frac{1}{\lambda_{\min}}\right| \leq \tilde{\mathcal{O}}\left(\frac{d}{n}\right)$$

where $\lambda_{\min}$ is the minimum over all eigenvalues of $W_i = \Sigma_i^{-1} - \Sigma_{i_*}^{-1}, \forall i \neq i_*$.

*Proof.* Let $\epsilon_\lambda = \hat{\lambda}_{\min} - \lambda_{\min}$ denote the error between the estimate and the true minimum eigenvalue. We have that

$$\left|\frac{1}{\hat{\lambda}_{\min}} - \frac{1}{\lambda_{\min}}\right| = \left|\frac{\lambda_{\min} - \hat{\lambda}_{\min}}{\lambda_{\min}\hat{\lambda}_{\min}}\right| = \left|\frac{\epsilon_\lambda}{\lambda_{\min}\hat{\lambda}_{\min}}\right| = \frac{|\epsilon_\lambda|}{\lambda_{\min}\left|\hat{\lambda}_{\min} - \lambda_{\min} + \lambda_{\min}\right|}$$

$$\leq \frac{|\epsilon_\lambda|}{\lambda_{\min}(\lambda_{\min} - |\epsilon_\lambda|)}$$

$$\leq \frac{|\epsilon_\lambda|}{\lambda_{\min}^2(1 - \frac{\epsilon_\lambda}{\lambda_{\min}})} \tag{61}$$

In order to bound the term $(1 - \epsilon_\lambda/\lambda_{\min})^{-1}$, we use the Taylor expansion of the function $f(x) = (1 - \frac{1}{x})^{-1} = \sum_{n=0}^{+\infty} x^n$ and get for $\epsilon_\lambda < \lambda_{\min}$ that

$$(1 - \epsilon_c/\lambda_{\min})^{-1} \leq \sum_{n=0}^{+\infty} \left(\frac{\epsilon_\lambda}{\lambda_{\min}}\right)^n = \tilde{\mathcal{O}}\left(\epsilon_\lambda\right) \tag{62}$$

Substituting inequality (62) into (61) and using from Lemma 11.6 the fact that $\epsilon_\lambda = \tilde{\mathcal{O}}\left(\sqrt{\frac{d}{n}}\right)$, we obtain

$$\left|\frac{1}{\hat{\lambda}_{\min}} - \frac{1}{\lambda_{\min}}\right| \leq \tilde{\mathcal{O}}\left(\frac{d}{n}\right)$$

$\square$

**Lemma 11.8.** For a sample $(x, y)$, let

$$c_M(x) = \max_{i \neq y} \|\Sigma_y^{-1}(x - \mu_y)^T - \Sigma_i^{-1}(x - \mu_i)^T\|_2$$

$$\hat{c}_M(x) = \max_{i \neq y} \|\hat{\Sigma}_y^{-1}(x - \hat{\mu}_y)^T - \hat{\Sigma}_i^{-1}(x - \hat{\mu}_i)^T\|_2$$

and $\hat{\mu}_j, \hat{\Sigma}_j$ the empirical mean and covariance of the Gaussian marginal $\mathcal{D}_j, \forall j \in [K]$. If the number of samples observed from each marginal $\mathcal{D}_j, \forall j \in [K]$ is $n \geq d$ and $\|\Sigma_j - \hat{\Sigma}_j\|_{\text{op}} < \frac{1}{\|\Sigma_i^{-1}\|_{\text{op}}}$, then we have that

$$|c_M - \hat{c}_M| \leq \tilde{\mathcal{O}}\left(\frac{d}{n}\right)$$

*Proof.* Let $\alpha_i = \Sigma_i^{-1}(x - \mu_i)^T, \forall i \in [K]$ and $\hat{\alpha}_i = \hat{\Sigma}_i^{-1}(x - \hat{\mu}_i)^T, \forall i \in [K]$. Then, we have that

$$c_M(x) = \max_{i \neq y} \|\alpha_y - \alpha_i\|_2, \quad \text{and} \quad \hat{c}_M(x) = \max_{i \neq y} \|\hat{\alpha}_y - \hat{\alpha}_i\|_2$$

Using the fact that for any two arbitrary sequences of real numbers $b_i, c_i$ it holds that

$$\left|\max_i b_i - \max_i c_i\right| \leq \max_i |b_i - c_i|$$

we have for $b_i = \|\alpha_y - \alpha_i\|_2$ and $c_i = \|\hat{\alpha}_y - \hat{\alpha}_i\|_2$ that

$$|c_M - \hat{c}_M| = \left|\max_{i \neq y} \|\alpha_y - \alpha_i\|_2 - \max_{i \neq y} \|\hat{\alpha}_y - \hat{\alpha}_i\|_2\right| \leq \max_{i \neq y} |\|\alpha_y - \alpha_i\|_2 - \|\hat{\alpha}_y - \hat{\alpha}_i\|_2| \tag{63}$$

Applying the triangle inequality on the norm $\|\alpha_y - \alpha_i\|_2 - \|\hat{\alpha}_y - \hat{\alpha}_i\|_2$, we have that $\|\alpha_y - \alpha_i\|_2 - \|\hat{\alpha}_y - \hat{\alpha}_i\|_2 \leq \|\alpha_y - \hat{\alpha}_y + \hat{\alpha}_i - \alpha_i\|_2$ we obtain

$$|c_M - \hat{c}_M| \leq \max_{i \neq y} |\|\alpha_y - \alpha_i\|_2 - \|\hat{\alpha}_y - \hat{\alpha}_i\|_2| \leq \max_{i \neq y} \|\alpha_y - \hat{\alpha}_y\|_2 + \|\alpha_i - \hat{\alpha}_i\|_2 \tag{64}$$

We, next, bound the error on the terms $\|\alpha_y - \hat{\alpha}_y\|_2$ and $\alpha_i - \hat{\alpha}_i$. It holds that

$$\begin{aligned}
\alpha_i - \hat{\alpha}_i &= \Sigma_i^{-1}(x - \mu_i)^T - \hat{\Sigma}_i^{-1}(x - \hat{\mu}_i)^T \\
&= \Sigma_i^{-1}(x - \mu_i)^T - \Sigma_i^{-1}(x - \hat{\mu}_i)^T + \Sigma_i^{-1}(x - \hat{\mu}_i)^T - \hat{\Sigma}_i^{-1}(x - \hat{\mu}_i)^T \\
&= \Sigma_i^{-1}(\hat{\mu}_i - \mu_i)^T + (\Sigma_i^{-1} - \hat{\Sigma}_i^{-1})(x - \hat{\mu}_i)^T \\
&= \Sigma_i^{-1}(\hat{\mu}_i - \mu_i)^T + (\Sigma_i^{-1} - \hat{\Sigma}_i^{-1})(\mu_i - \hat{\mu}_i)^T + (\Sigma_i^{-1} - \hat{\Sigma}_i^{-1})(x - \mu_i)^T
\end{aligned}$$

Taking the norm and applying the triangle inequality, we get

$$\begin{aligned}
\|\alpha_i - \hat{\alpha}_i\|_2 &\leq \|\Sigma_i^{-1}(\hat{\mu}_i - \mu_i)^T\|_2 + \|(\Sigma_i^{-1} - \hat{\Sigma}_i^{-1})(\mu_i - \hat{\mu}_i)^T\|_2 + \|(\Sigma_i^{-1} - \hat{\Sigma}_i^{-1})(x - \mu_i)^T\|_2 \\
&\leq \|\Sigma_i^{-1}\|_{\text{op}}\|\hat{\mu}_i - \mu_i\|_2 + \|\Sigma_i^{-1} - \hat{\Sigma}_i^{-1}\|_{\text{op}}\|\mu_i - \hat{\mu}_i\|_2 + \|\Sigma_i^{-1} - \hat{\Sigma}_i^{-1}\|_{\text{op}}\|x - \mu_i\|_2 \tag{65}
\end{aligned}$$

In order to bound inequality (65), we need upper bounds on $\|\mu_i - \hat{\mu}_i\|_2$ and $\|\Sigma_i^{-1} - \hat{\Sigma}_i^{-1}\|_{\text{op}}$. We have that $\|\mu_i - \hat{\mu}_i\|_2 = \tilde{\mathcal{O}}\left(\sqrt{\frac{d}{n}}\right)$, where $n$ is the minimum number of samples observed from each marginal $\mathcal{D}_i, \forall i \in [K]$. For the term $\|\Sigma_i^{-1} - \hat{\Sigma}_i^{-1}\|_{\text{op}}$, we get for $\|\Sigma_i - \hat{\Sigma}_i\|_{\text{op}} < \frac{1}{\|\Sigma_i^{-1}\|_{\text{op}}}$ that the following inequality holds

$$\|\Sigma_i^{-1} - \hat{\Sigma}_i^{-1}\|_{\text{op}} \le \|\Sigma_i^{-1}\|_{\text{op}}^2 \|\Sigma_i - \hat{\Sigma}_i\|_{\text{op}}$$

Using the fact that $\|\Sigma_i - \hat{\Sigma}_i\|_{\text{op}} = \tilde{\mathcal{O}}\left(\sqrt{\frac{d}{n}}\right)$ we obtain

$$\|\Sigma_i^{-1} - \hat{\Sigma}_i^{-1}\|_{\text{op}} \le \tilde{\mathcal{O}}\left(\sqrt{\frac{d}{n}}\right) \tag{66}$$

Substituting inequality (66) and the fact that $\|\mu_i - \hat{\mu}_i\|_2 = \tilde{\mathcal{O}}\left(\sqrt{\frac{d}{n}}\right)$ into (65)

$$\|\alpha_i - \hat{\alpha}_i\|_2 \quad \le \quad \|\Sigma_i^{-1}\|_{\text{op}} \tilde{\mathcal{O}}\left(\sqrt{\frac{d}{n}}\right) + \tilde{\mathcal{O}}\left(\frac{d}{n}\right) + \tilde{\mathcal{O}}\left(\sqrt{\frac{d}{n}}\right) \le \tilde{\mathcal{O}}\left(\frac{d}{n}\right) \tag{67}$$

Similarly, for $i = y$ we have that

$$\|\alpha_y - \hat{\alpha}_y\|_2 \quad \le \quad \tilde{\mathcal{O}}\left(\frac{d}{n}\right) \tag{68}$$

From (64), (67), (68), we get that

$$|c_M - \hat{c}_M| \quad \le \quad \|\alpha_y - \hat{\alpha}_y\|_2 + \max_{i \ne y} \|\alpha_i - \hat{\alpha}_i\|_2 \le \tilde{\mathcal{O}}\left(\frac{d}{n}\right)$$

$\square$

**Lemma 11.9.** Let $A = \sqrt{c_M^2 - m(x)\lambda_{\min}}$ and $\hat{A} = \sqrt{\hat{c}_M^2 - \hat{m}(x)\hat{\lambda}_{\min}}$. If the minimum number of samples observed from each marginal $\mathcal{D}_j, \forall j \in [K]$ is $n \ge d$ and $\|\Sigma_j - \hat{\Sigma}_j\|_{\text{op}} < \frac{1}{\|\Sigma_j^{-1}\|_{\text{op}}}$, then we have that

$$|A - \hat{A}| \quad \le \quad \tilde{\mathcal{O}}\left(\frac{d}{n}\right)$$

*Proof.* Using the Holder's inequality for $f(x) = \sqrt{x}$, we have that

$$\left|A - \hat{A}\right| \quad = \quad \left|\sqrt{c_M^2 - m(x)\lambda_{\min}} - \sqrt{\hat{c}_M^2 - \hat{m}(x)\hat{\lambda}_{\min}}\right|$$

$$\le \quad \sqrt{\left|c_M^2 - \hat{c}_M^2 - m(x)\lambda_{\min} + \hat{m}(x)\hat{\lambda}_{\min}\right|} \tag{69}$$

We, now, express the quantity under the root with respect to $\epsilon_c = \hat{c}_M - c_M$ and $\epsilon_\lambda = \hat{\lambda}_{min} - \lambda_{min}$, as follows

$$
\begin{aligned}
c_M^2 - \hat{c}_M^2 - m(x)\lambda_{\min} + \hat{m}(x)\hat{\lambda}_{\min} \quad &\le \quad (c_M - \hat{c}_M)(c_M + \hat{c}_M) - m(x)\lambda_{\min} + \hat{m}(x)\hat{\lambda}_{\min} \\
&= \quad \epsilon_c(2c_M + \epsilon) - m(x)(\lambda_{\min} - \hat{\lambda}_{\min}) + \hat{\lambda}_{\min}(m(x) - \hat{m}(x)) \\
&= \quad \epsilon_c(2c_M + \epsilon) - m(x)(\lambda_{\min} - \hat{\lambda}_{\min}) \\
&\quad + (\hat{\lambda}_{\min} - \lambda_{\min})(m(x) - \hat{m}(x)) + \lambda_{\min}(m(x) - \hat{m}(x)) \tag{70}
\end{aligned}
$$

Combining (69), (70), using triangle inequality and the concavity of the square root, we get

$$
\begin{aligned}
\left|A - \hat{A}\right| \quad &\le \quad \sqrt{\epsilon_c(2c_M + \epsilon_c)} + \sqrt{m(x)\left|\lambda_{\min} - \hat{\lambda}_{\min}\right|} + \sqrt{\left|(\hat{\lambda}_{\min} - \lambda_{\min})(m(x) - \hat{m}(x))\right|} \\
&\quad + \sqrt{\left|\lambda_{\min}(m(x) - \hat{m}(x))\right|} \\
&\le \quad \sqrt{\epsilon_c(2c_M + \epsilon_c)} + \sqrt{m(x)\epsilon_\lambda} + \sqrt{\epsilon_\lambda \epsilon_m} + \sqrt{|\lambda_{\min}\epsilon_m|} \tag{71}
\end{aligned}
$$

From Lemma 11.6, 11.8, 11.5, we have that $\epsilon_m = \tilde{\mathcal{O}}\left(\frac{d^{3/2}}{n^{3/2}}\right), \epsilon_c = \tilde{\mathcal{O}}\left(\frac{d}{n}\right)$ and $\epsilon_\lambda = \tilde{\mathcal{O}}\left(\sqrt{\frac{d}{n}}\right)$ and thus

$$\left| A - \hat{A} \right| \leq \tilde{\mathcal{O}}\left(\frac{d}{n}\right) + \tilde{\mathcal{O}}\left(\frac{d^{1/4}}{n^{1/4}}\right) + \tilde{\mathcal{O}}\left(\frac{d}{n}\right) + \tilde{\mathcal{O}}\left(\frac{d^{3/4}}{n^{3/4}}\right) = \tilde{\mathcal{O}}\left(\frac{d}{n}\right)$$

$\square$

**Lemma 11.10.** For an input sample $(x, y)$, with $|\hat{c}_M - c_M| \leq c_M$ and number of observed samples from each marginal $\mathcal{D}_j, \forall j \in [K]$ at least $n \geq d$, it holds that

$$\left| \frac{m(x)}{2c_M} - \frac{\hat{m}(x)}{2\hat{c}_M} \right| \leq \tilde{\mathcal{O}}\left(\frac{d^{3/2}}{n^{3/2}}\right)$$

*Proof.* From triangle inequality, we have that

$$\left| \frac{m(x)}{2c_M} - \frac{\hat{m}(x)}{2\hat{c}_M} \right| = \left| \frac{m(x)}{2c_M} - \frac{\hat{m}(x)}{2c_M} + \frac{\hat{m}(x)}{2c_M} - \frac{\hat{m}(x)}{2\hat{c}_M} \right|$$

$$\leq \frac{1}{2c_M} |m(x) - \hat{m}(x)| + \frac{\hat{m}(x)}{2} \left| \frac{1}{c_M} - \frac{1}{\hat{c}_M} \right|$$

$$\leq \frac{|m(x) - \hat{m}(x)|}{2c_M} + \left( \frac{m(x)}{2} + \frac{|m(x) - \hat{m}(x)|}{2} \right) \left| \frac{1}{c_M} - \frac{1}{\hat{c}_M} \right|$$

$$= \frac{|m(x) - \hat{m}(x)|}{2c_M} + \left( \frac{m(x)}{2c_M} + \frac{|m(x) - \hat{m}(x)|}{2c_M} \right) \left| \frac{\hat{c}_M - c_M}{\hat{c}_M} \right| \quad (72)$$

We, next, bound the terms $|m(x) - \hat{m}(x)|$ and $\left| \frac{\hat{c}_M - c_M}{\hat{c}_M} \right|$. From Lemma 11.5, we have that

$$|\hat{m}(x) - m(x)| \leq \tilde{\mathcal{O}}\left(\frac{d^{3/2}}{n^{3/2}}\right) \quad (73)$$

From Lemma 11.8, we have that $|\hat{c}_M - c_M| \leq \epsilon_c$ with $\epsilon_c = \tilde{\mathcal{O}}\left(\frac{d}{n}\right)$ and assuming that $\epsilon_c < c_M$, we have that

$$\left| \frac{\hat{c}_M - c_M}{\hat{c}_M} \right| \leq \frac{\epsilon_c}{|\hat{c}_M|} \leq \frac{\epsilon_c}{c_M - |\hat{c}_M - c_M|} \leq \frac{\epsilon_c}{c_M - \epsilon_c} = \frac{\epsilon_c}{c_M(1 - \epsilon_c/c_M)} \quad (74)$$

In order to bound the term $(1 - \epsilon_c/c_M)^{-1}$, we use the Taylor expansion of the function $f(x) = (1 - \frac{1}{x})^{-1} = \sum_{n=0}^{+\infty} x^n$ and get for $\epsilon_c < c_M$ that

$$(1 - \epsilon_c/c_M)^{-1} \leq \sum_{n=0}^{+\infty} \left( \frac{\epsilon_c}{c_M} \right)^n = \tilde{\mathcal{O}}(\epsilon_c) = \tilde{\mathcal{O}}\left(\frac{d}{n}\right) \quad (75)$$

Substituting (75) into (74), we get that

$$\left| \frac{\hat{c}_M - c_M}{\hat{c}_M} \right| \leq \tilde{\mathcal{O}}\left(\epsilon_c \frac{d}{n}\right) \leq \tilde{\mathcal{O}}\left(\frac{d^2}{n^2}\right) \quad (76)$$

Combining (73), (76) with (72), we obtain

$$\left| \frac{m(x)}{2c_M} - \frac{\hat{m}(x)}{2\hat{c}_M} \right| \leq \tilde{\mathcal{O}}\left(\frac{d^{3/2}}{n^{3/2}}\right) + \left[ \frac{m(x)}{2c_M} + \tilde{\mathcal{O}}\left(\frac{d^{3/2}}{n^{3/2}}\right) \right] \tilde{\mathcal{O}}\left(\frac{d^2}{n^2}\right) \leq \tilde{\mathcal{O}}\left(\frac{d^{3/2}}{n^{3/2}}\right) \quad (77)$$

$\square$

**Lemma 11.11.** For an input sample $(x, y)$ with $|\hat{\lambda}_{\min} - \lambda_{\min}| < \lambda_{\min}, \|\Sigma_j - \hat{\Sigma}_j\|_{\text{op}} < \frac{1}{\|\Sigma_j^{-1}\|_{\text{op}}}$, and number of observed samples from each marginal $\mathcal{D}_j, \forall j \in [K]$ at least $n \geq d$, then it holds

that

$$\left| \frac{\hat{c}_M - \sqrt{\hat{c}_M^2 - \hat{m}(x)\hat{\lambda}_{min}}}{\hat{\lambda}_{min}} - \frac{\hat{m}(x)}{2\hat{c}_M} \right| \leq \tilde{\mathcal{O}}\left( \frac{d^{9/2}}{n^{9/2}} \right)$$

*Proof.* We use the Taylor expansion of the function $f(x) = \sqrt{\hat{c}_M^2 - x} = \hat{c}_M \sum_{n=0}^{\infty} \binom{1/2}{n}(-1)^n \left( \frac{x}{\hat{c}_M^2} \right)^n$ and get

$$\sqrt{\hat{c}_M^2 - \hat{m}(x)\hat{\lambda}_{min}} = \hat{c}_M \sum_{n=0}^{\infty} \binom{1/2}{n}(-1)^n \left( \frac{\hat{m}(x)\hat{\lambda}_{min}}{\hat{c}_M^2} \right)^n$$

$$\Rightarrow \left| \frac{\hat{c}_M - \sqrt{\hat{c}_M^2 - \hat{m}(x)\hat{\lambda}_{min}}}{\hat{\lambda}_{min}} - \frac{\hat{m}(x)}{2\hat{c}_M} \right| \leq \tilde{\mathcal{O}}\left( \frac{\hat{m}^2(x)\hat{\lambda}_{min}}{8\hat{c}_M^3} \right) \tag{78}$$

Let $\epsilon_m = \hat{m}(x) - m(x), \epsilon_c = \hat{c}_M - c_M$ and $\epsilon_\lambda = \hat{\lambda}_{min} - \lambda_{min}$. Then, we have that

$$\frac{\hat{m}^2(x)\hat{\lambda}_{min}}{8\hat{c}_M^3} = \frac{(m(x) + \epsilon_m)^2(\lambda_{min} + \epsilon_\lambda)}{8(c_M + \epsilon_c)^3} = \frac{(m^2(x) + 2\epsilon_m m(x) + \epsilon_m^2)(\lambda_{min} + \epsilon_\lambda)}{8c_M^3(1 + \epsilon_c/c_M)^3} \tag{79}$$

Using the Taylor expansion of $f(x) = (1 + \frac{1}{x})^{-3} = \sum_{n=0}^{\infty}(-1)^n \frac{(n+2)(n+1)}{2} x^{-n}$, we have for $\epsilon_c < c_M$ that $(1 + \epsilon_c/c_M)^{-3} = 1 - 3\frac{\epsilon_c}{c_M} + 6(\frac{\epsilon_c}{c_M})^2 + ... \leq \tilde{\mathcal{O}}(\epsilon_c)$ and thus

$$\frac{\hat{m}^2(x)\hat{\lambda}_{min}}{8\hat{c}_M^3} \leq \frac{(m^2(x) + 2\epsilon_m m(x) + \epsilon_m^2)(\lambda_{min} + \epsilon_\lambda)}{8c_M} \tilde{\mathcal{O}}(\epsilon_c) = \tilde{\mathcal{O}}\left( \epsilon_m^2 \epsilon_\lambda \epsilon_c \right)$$

From Lemma 11.6, 11.5, 11.8 we have that $\epsilon_m^2 = \tilde{\mathcal{O}}\left( \frac{d^3}{n^3} \right), \epsilon_c = \tilde{\mathcal{O}}\left( \frac{d}{n} \right)$ and $\epsilon_\lambda = \tilde{\mathcal{O}}\left( \sqrt{\frac{d}{n}} \right)$ and thus we obtain

$$\left| \frac{\hat{c}_M - \sqrt{\hat{c}_M^2 - \hat{m}(x)\hat{\lambda}_{min}}}{\hat{\lambda}_{min}} - \frac{\hat{m}(x)}{2\hat{c}_M} \right| \leq \tilde{\mathcal{O}}\left( \frac{d^{9/2}}{n^{9/2}} \right) \tag{80}$$

$\square$

**Lemma 11.12.** For an input sample $(x, y)$ with $|\hat{\lambda}_{min} - \lambda_{min}| < \lambda_{min}, \|\Sigma_j - \hat{\Sigma}_j\|_{op} < \frac{1}{\|\Sigma_j^{-1}\|_{op}}, |\hat{c}_M - c_M| < c_M$ and number of observed samples from each marginal $\mathcal{D}_j, \forall j \in [K]$ at least $n \geq d$, it holds that

$$\left| \frac{\hat{c}_M - \sqrt{\hat{c}_M^2 - \hat{m}(x)\hat{\lambda}_{min}}}{\hat{\lambda}_{min}} - \frac{m(x)}{2c_M} \right| \leq \tilde{\mathcal{O}}\left( \frac{d^{9/2}}{n^{9/2}} \right)$$

*Proof.* Applying the triangle inequality, we have that

$$\left| \frac{\hat{c}_M - \sqrt{\hat{c}_M^2 - \hat{m}(x)\hat{\lambda}_{min}}}{\hat{\lambda}_{min}} - \frac{m(x)}{2c_M} \right| \leq \underbrace{\left| \frac{\hat{c}_M - \sqrt{\hat{c}_M^2 - \hat{m}(x)\hat{\lambda}_{min}}}{\hat{\lambda}_{min}} - \frac{\hat{m}(x)}{2\hat{c}_M} \right|}_{T_1}$$

$$+ \underbrace{\left| \frac{\hat{m}(x)}{2\hat{c}_M} - \frac{m(x)}{2c_M} \right|}_{T_2} \tag{81}$$

We, next, bound the terms $T_1, T_2$ appearing on the right-hand side of (81). From Lemma 11.11, we have for $\epsilon_c < c_M$ that

$$T_1 \leq \tilde{\mathcal{O}}\left( \frac{d^{9/2}}{n^{9/2}} \right) \tag{82}$$

From Lemma 11.10, we have that

$$T_2 \leq \tilde{\mathcal{O}}\left(\frac{d^{3/2}}{n^{3/2}}\right) \tag{83}$$

Substituting (82), (83) to (81), we get that

$$\left| \frac{\hat{c}_M - \sqrt{\hat{c}_M^2 - \hat{m}(x)\hat{\lambda}_{min}}}{\hat{\lambda}_{min}} - \frac{m(x)}{2c_M} \right| \leq \tilde{\mathcal{O}}\left(\frac{d^{9/2}}{n^{9/2}}\right) \tag{84}$$

$\square$

**Lemma 11.13.** For an input sample $(x, y)$ with $\hat{\lambda}_{min} - \lambda_{min} < \lambda_{min}, |\hat{c}_M - c_M| < c_M$ and number of observed samples from each marginal $\mathcal{D}_j, \forall j \in [K]$ at least $n \geq d$, it holds that

$$\left| \frac{c_M - \sqrt{c_M^2 - m(x)\lambda_{min}}}{\lambda_{min}} - \frac{\hat{c}_M - \sqrt{\hat{c}_M^2 - \hat{m}(x)\hat{\lambda}_{min}}}{\hat{\lambda}_{min}} \right| \leq \tilde{\mathcal{O}}\left(\frac{d^{3/2}}{n^{3/2}}\right)$$

*Proof.* Let $\mathcal{R}(x), \hat{\mathcal{R}}(x)$ be the true and learnt certificate of robustness from Theorem 3.1. We have that

$$|\hat{\mathcal{R}}(x) - \mathcal{R}(x)| = \left| \frac{c_M - \sqrt{c_M^2 - m(x)\lambda_{min}}}{\lambda_{min}} - \frac{\hat{c}_M - \sqrt{\hat{c}_M^2 - \hat{m}(x)\hat{\lambda}_{min}}}{\hat{\lambda}_{min}} \right|$$

$$= \left| \frac{c_M - \sqrt{c_M^2 - m(x)\lambda_{min}}}{\lambda_{min}} - \frac{c_M - \sqrt{c_M^2 - m(x)\lambda_{min}}}{\hat{\lambda}_{min}} \right.$$

$$\left. + \frac{c_M - \sqrt{c_M^2 - m(x)\lambda_{min}}}{\hat{\lambda}_{min}} - \frac{\hat{c}_M - \sqrt{\hat{c}_M^2 - \hat{m}(x)\hat{\lambda}_{min}}}{\hat{\lambda}_{min}} \right|$$

Let $A = \sqrt{c_M^2 - m(x)\lambda_{min}}$ and $\hat{A} = \sqrt{\hat{c}_M^2 - \hat{m}(x)\hat{\lambda}_{min}}$. By applying the triangle inequality, we get

$$|\hat{\mathcal{R}}(x) - \mathcal{R}(x)| \leq \left| \frac{c_M - A}{\lambda_{min}} - \frac{c_M - A}{\hat{\lambda}_{min}} \right| + \left| \frac{c_M - A}{\hat{\lambda}_{min}} - \frac{\hat{c}_M - \hat{A}}{\hat{\lambda}_{min}} \right|$$

$$\leq \left| \frac{c_M - A}{\lambda_{min}} - \frac{c_M - A}{\hat{\lambda}_{min}} \right| + \left| \frac{A - \hat{A}}{\hat{\lambda}_{min}} \right| + \left| \frac{c_M - \hat{c}_M}{\hat{\lambda}_{min}} \right|$$

$$= |c_M - A| \left| \frac{1}{\lambda_{min}} - \frac{1}{\hat{\lambda}_{min}} \right| + \left| \frac{1}{\hat{\lambda}_{min}} \right| \left( \left| A - \hat{A} \right| + |c_M - \hat{c}_M| \right)$$

$$= |c_M - A| \left| \frac{1}{\lambda_{min}} - \frac{1}{\hat{\lambda}_{min}} \right| + \left| \frac{1}{\hat{\lambda}_{min}} - \frac{1}{\lambda_{min}} \right| \left( \left| A - \hat{A} \right| + |c_M - \hat{c}_M| \right)$$

$$+ \left| \frac{1}{\lambda_{min}} \right| \left( \left| A - \hat{A} \right| + |c_M - \hat{c}_M| \right) \tag{85}$$

Thus, in order to bound inequality (85) we need to bound the terms $\left| \frac{1}{\lambda_{min}} - \frac{1}{\hat{\lambda}_{min}} \right|, \left| A - \hat{A} \right|$ and $|c_M - \hat{c}_M|$. From Lemmas 11.7, 11.8, 11.9, we have that for $\epsilon_\lambda = \hat{\lambda}_{min} - \lambda_{min} \leq \lambda_{min}, \|\Sigma_i - \hat{\Sigma}_i\|_{op} < \frac{1}{\|\Sigma_i^{-1}\|_{op}}$, it holds that

$$\left| \frac{1}{\lambda_{min}} - \frac{1}{\hat{\lambda}_{min}} \right| \leq \tilde{\mathcal{O}}\left(\frac{d}{n}\right) \tag{86}$$

$$\left| A - \hat{A} \right| \leq \tilde{\mathcal{O}}\left(\frac{d}{n}\right) \tag{87}$$

$$|c_M - \hat{c}_M| \leq \tilde{\mathcal{O}}\left(\frac{d}{n}\right) \tag{88}$$

Substituting inequalities (86), (87), (88) into (85), we get

$$
\begin{aligned}
|\hat{\mathcal{R}}(x) - \mathcal{R}(x)| &\leq |c_M - A| \left| \frac{1}{\lambda_{\min}} - \frac{1}{\hat{\lambda}_{\min}} \right| + \left| \frac{1}{\hat{\lambda}_{\min}} - \frac{1}{\lambda_{\min}} \right| \left( \left| A - \hat{A} \right| + |c_M - \hat{c}_M| \right) \\
&\quad + \left| \frac{1}{\lambda_{\min}} \right| \left( \left| A - \hat{A} \right| + |c_M - \hat{c}_M| \right) \\
&\leq |c_M - A| \, \tilde{\mathcal{O}}\left( \sqrt{\frac{d}{n}} \right) + \tilde{\mathcal{O}}\left( \sqrt{\frac{d}{n}} \right) \tilde{\mathcal{O}}\left( \frac{d^2}{n^2} \right) + \left| \frac{1}{\lambda_{\min}} \right| \tilde{\mathcal{O}}\left( \frac{d}{n} \right) \\
&\leq \tilde{\mathcal{O}}\left( \frac{d^{3/2}}{n^{3/2}} \right)
\end{aligned}
$$

$\square$

**Lemma 11.14.** If $\hat{\lambda}_{\min} - \lambda_{\min} < \lambda_{\min}, |\hat{c}_M - c_M| < c_M, \|\Sigma_j - \hat{\Sigma}_j\|_{\text{op}} < \frac{1}{\|\Sigma_j^{-1}\|_{\text{op}}}$,, the following bound holds

$$
\left| \frac{c_M - \sqrt{c_M^2 - m(x)\lambda_{\min}}}{\lambda_{\min}} - \frac{\hat{m}(x)}{2\hat{c}_M} \right| \leq \tilde{\mathcal{O}}\left( \frac{d^{9/2}}{n^{9/2}} \right)
$$

*Proof.* From triangle inequality, we have that

$$
\left| \frac{c_M - \sqrt{c_M^2 - m(x)\lambda_{\min}}}{\lambda_{\min}} - \frac{\hat{m}(x)}{2\hat{c}_M} \right| \leq \underbrace{\left| \frac{c_M - \sqrt{c_M^2 - m(x)\lambda_{\min}}}{\lambda_{\min}} - \frac{\hat{c}_M - \sqrt{\hat{c}_M^2 - \hat{m}(x)\hat{\lambda}_{min}}}{\hat{\lambda}_{min}} \right|}_{T_1}
$$
$$
+ \underbrace{\left| \frac{\hat{c}_M - \sqrt{\hat{c}_M^2 - \hat{m}(x)\hat{\lambda}_{min}}}{\hat{\lambda}_{min}} - \frac{\hat{m}(x)}{2\hat{c}_M} \right|}_{T_2} \tag{89}
$$

From Lemma 11.13, Lemma 11.11, we have for $\epsilon_c = |\hat{c}_M - c_M| < c_M$ that

$$
T_1 \leq \tilde{\mathcal{O}}\left( \frac{d^{3/2}}{n^{3/2}} \right) \tag{90}
$$

$$
T_2 \leq \tilde{\mathcal{O}}\left( \frac{d^{9/2}}{n^{9/2}} \right) \tag{91}
$$

Substituting (90), (91) into (89), we obtain

$$
\left| \frac{c_M - \sqrt{c_M^2 - m(x)\lambda_{\min}}}{\lambda_{\min}} - \frac{\hat{m}(x)}{2\hat{c}_M} \right| \leq \tilde{\mathcal{O}}\left( \frac{d^{9/2}}{n^{9/2}} \right)
$$

$\square$

## 11.2 PROOF OF THEOREM 3.2.

*Proof.* Let $\hat{\mu}_i, \hat{\Sigma}_i, \hat{\pi}_i$ be the learnt parameters and $\mu_i, \Sigma_i, \pi_i$ the true parameters of the Gaussian marginal $\mathcal{D}_i, \forall i \in [K]$. Denote with $n_i$ the number of samples observed from $\mathcal{D}_i$ and let $n = \min_{i \in [K]} n_i$ be the minimum number of samples observed from any marginal distribution. Using Gaussian concentration results (Theorem 6.1 Wai (2019)), we have that the empirical mean $\hat{\mu}_i$ and empirical covariance $\hat{\Sigma}_i$ of each Gaussian marginal $\mathcal{D}_i, \forall i \in [K]$, satisfy

$$
\|\hat{\mu}_i - \mu_i\|_{\Sigma^{-1}} \leq \tilde{\mathcal{O}}\left( \sqrt{\frac{d}{n}} \right), \quad \|\hat{\Sigma} - \Sigma_i\|_{\text{op}} \leq \tilde{\mathcal{O}}\left( \sqrt{\frac{d}{n}} \right)
$$

where $\tilde{\mathcal{O}}(\cdot)$ suppresses logarithmic terms in $\delta$. For samples $x_1, x_2, ..., x_n \sim$ Multinomial$(\pi_1, ..., \pi_K)$ and $\hat{\pi}_i = \frac{1}{n} \sum_{j=1}^{n} \mathbb{I}\{y_j = i\}$, from the Hoeffding's bound we get that with probability at least $1 - \delta$, it holds

$$\|\hat{\pi}_i - \pi_i\|_2 \leq \tilde{\mathcal{O}}\left(\sqrt{\frac{1}{n}}\right), \quad \forall i \in [K],$$

Given the above bounds on the distance of the estimated parameters from the true ones, we bound the learnt certificate of robustness $\hat{\mathcal{R}}(x)$ from the true certificate $\mathcal{R}(x)$. From (31), (32) in Theorem 3.1, depending on the sign of $\lambda_{\min}$, there are two cases for the expression of the certified radius, specifically $\mathcal{R}(x) = \frac{c_M - \sqrt{c_M^2 - m(x)\lambda_{\min}}}{\lambda_{\min}}$ or $\mathcal{R}(x) = \frac{m(x)}{2c_M}$. Similarly, based on whether $\hat{\lambda}_{min}$ is positive or negative, there are two cases for the expression of $\hat{\mathcal{R}}(x)$.

We, thus, partition the input space $\mathcal{X}$ into four disjoint subspaces $\mathcal{X}_1, \mathcal{X}_2, \mathcal{X}_3, \mathcal{X}_4$, where

$$\mathcal{X}_1 = \{x \in \mathcal{X} : \lambda_{\min} > 0, \hat{\lambda}_{\min} > 0\}$$
$$\mathcal{X}_2 = \{x \in \mathcal{X} : \lambda_{\min} > 0, \hat{\lambda}_{\min} \leq 0\}$$
$$\mathcal{X}_3 = \{x \in \mathcal{X} : \lambda_{\min} \leq 0, \hat{\lambda}_{\min} > 0\}$$
$$\mathcal{X}_4 = \{x \in \mathcal{X} : \lambda_{\min} \leq 0, \hat{\lambda}_{\min} \leq 0\}$$

Based on the above partition of $\mathcal{X}$, we have that

$$\begin{aligned}
\mathbb{P}_{x \sim \mathcal{D}}\left[|\mathcal{R}(x) - \hat{\mathcal{R}}(x)| \geq \epsilon\right] &= \sum_{i \in [4]} \mathbb{P}_{x \sim \mathcal{D}}[x \in \mathcal{X}_i] \, \mathbb{P}\left[|\mathcal{R}(x) - \hat{\mathcal{R}}(x)| \geq \epsilon_i \Big| x \in \mathcal{X}_i\right] \\
&\leq \sum_{i \in [4]} \mathbb{P}_{x \sim \mathcal{D}}[x \in \mathcal{X}_i] \, \delta \\
&\leq \delta
\end{aligned} \tag{92}$$

where $\delta = \mathbb{P}\left[|\mathcal{R}(x) - \hat{\mathcal{R}}(x)| \geq \epsilon_i \Big| x \in \mathcal{X}_i\right]$. To prove the needed, thus, it suffices to fix a probability $\delta$ and find the errors $\epsilon_i$ for each of the four cases, and, finally, let $\epsilon = \max_{i \in [4]} \epsilon_i$ in (92).

**Case 1 ($\mathcal{X}_1$).** We have that $\mathcal{R}(x) = \frac{m(x)}{2c_M}$ and $\hat{\mathcal{R}}(x) = \frac{\hat{m}(x)}{2\hat{c}_M}$ and thus according to Lemma 11.10 for $\epsilon < c_M$, it holds that

$$|\mathcal{R}(x) - \hat{\mathcal{R}}(x)| = \tilde{\mathcal{O}}\left(\frac{d^{3/2}}{n^{3/2}}\right)$$

**Case 2 ($\mathcal{X}_2$).** We have that $\mathcal{R}(x) = \frac{m(x)}{2c_M}$ and $\hat{\mathcal{R}}(x) = \frac{\hat{c}_M - \sqrt{\hat{c}_M^2 - \hat{m}(x)\hat{\lambda}_{min}}}{\hat{\lambda}_{min}}$ and according to Lemma 11.12 for $\epsilon < \min\{\lambda_{min}, \lambda_{min}(\Sigma_i), c_M\}$, it holds that

$$|\mathcal{R}(x) - \hat{\mathcal{R}}(x)| = \tilde{\mathcal{O}}\left(\frac{d^{9/2}}{n^{9/2}}\right)$$

**Case 3 ($\mathcal{X}_3$).** We have that $\mathcal{R}(x) = \frac{c_M - \sqrt{c_M^2 - m(x)\lambda_{\min}}}{\lambda_{\min}}$ and $\hat{\mathcal{R}}(x) = \frac{\hat{m}(x)}{2\hat{c}_M}$ and according to Lemma 11.14 for $\epsilon < \min\{\lambda_{min}, \lambda_{min}(\Sigma_i), c_M\}$, it holds that

$$|\mathcal{R}(x) - \hat{\mathcal{R}}(x)| = \tilde{\mathcal{O}}\left(\frac{d^{9/2}}{n^{9/2}}\right)$$

**Case 4 ($\mathcal{X}_4$).** We have that $\mathcal{R}(x) = \frac{c_M - \sqrt{c_M^2 - m(x)\lambda_{\min}}}{\lambda_{\min}}$ and $\hat{\mathcal{R}}(x) = \frac{\hat{c}_M - \sqrt{\hat{c}_M^2 - \hat{m}(x)\hat{\lambda}_{min}}}{\hat{\lambda}_{min}}$ and thus according to Lemma 11.13 for $\epsilon < \min\{\lambda_{min}, c_M\}$, it holds that

$$|\mathcal{R}(x) - \hat{\mathcal{R}}(x)| = \tilde{\mathcal{O}}\left(\frac{d^{3/2}}{n^{3/2}}\right)$$

Combining the above results with (92), we get that with probability at least $1 - \delta$ it holds that

$$|\mathcal{R}(x) - \hat{\mathcal{R}}(x)| \leq \tilde{\mathcal{O}}\left(\frac{d^{9/2}}{n^{9/2}}\right)$$

For a fixed error $0 < \epsilon < \epsilon_{\min} = \{\lambda_{\min}(\Sigma_j), \lambda_{\min}, c_M(x)\}$, in order to satisfy that

$$|\hat{\mathcal{R}}(x) - \mathcal{R}(x)| < \mathcal{O}(\epsilon),$$

we have that number of samples needed are $n = \tilde{\mathcal{O}}\left(\frac{d^{9/2}}{\epsilon^{9/2}}\right)$ and this concludes the proof. $\qquad\square$

## 12 PROOF OF THEOREM 4.1

*Proof.* Denote with $x, x'$ the clean and the corresponding adversarially perturbed sample and with $z = f(x), z' = f(x')$ their embeddings in the latent space. Since the encoder network is $L$-Lipschitz, we have that

$$\|z - z'\|_2 = \|f(x) - f(x')\|_2 \leq L\|x - x'\|_2 \tag{93}$$

Given that $f(x)$ maps the input distribution to a latent distribution that is a mixture of Gaussians, we have from Theorem 3.1 that the ELLIPS classifier remains robust as long as

$$\|z' - z\|_2 \leq \frac{m(z)}{\lambda_{\min}\left(\sqrt{c_M^2 + (\lambda_{\min})_+ m(z)} + c_M\right)} \tag{94}$$

where $\lambda_{\min}$ is the minimum among all eigenvalues of the matrices $W_i = \Sigma_i^{-1} - \Sigma_{i_*}^{-1}, \forall i \neq i_*$, $(-\lambda_{\min})_+ = \max(-\lambda_{\min}, 0)$, and $c_M = \max_{i \neq i_*}\|\Sigma_{i_*}^{-1}(f(x) - \mu_{i_*})^T - \Sigma_i^{-1}(f(x) - \mu_i)^T\|_2$. Thus, in order for the classifier to remain robust, it suffices that the perturbation in the input space satisfies

$$L\|x - x'\|_2 \leq \frac{m(z)}{\lambda_{\min}\left(\sqrt{c_M^2 + (\lambda_{\min})_+ m(z)} + c_M\right)}$$

$$\iff \|x - x'\|_2 \leq \frac{m(z)}{\lambda_{\min} L\left(\sqrt{c_M^2 + (\lambda_{\min})_+ m(z)} + c_M\right)}$$

thus concluding the proof. $\qquad\square$

## 13 ON EXPERIMENTS

In Appendix 13.1, we provide more details on the experiments presented in the main paper. In Appendix 13.2, we provide additional experiments showcasing the performance of the proposed classifier in practice.

### 13.1 EXPERIMENTAL DETAILS

We first describe the experimental setup used and then provide additional synthetic experiments.

**Experiments in Benchmark Datasets.** We provide the training details - network architecture, datasets, optimization and hyperparameters for the implementation of the GENELLIPS classifier.

**Network Architecture.** To construct the proposed classifier we need to apply first an encoder and then the ELLIPS classifier. We take a FARE-4 encoder (Schlarmann et al., 2024) pre-trained and finetune it using a loss that promotes the latent distribution to comprise a mixture of Gaussians. Given that the ImageNet dataset appears to have more classes and be more complex than the CIFAR-10, we have utilized a meticulously constructed loss accustomed to each dataset. Specifically, for CIFAR-10 the used loss combines the MCR$^2$ objective with a term promoting the Gaussian marginals to be isotropic, ensuring that the eigenvalues of the covariance matrices are well-behaved

$$\mathcal{L} = \mathcal{L}_{\mathrm{MCR}^2}(Z, Y) + \lambda_{\mathrm{iso}}\mathcal{L}_{\mathrm{iso}} \tag{95}$$

where $\mathcal{L}_{\text{iso}} = \sum_{k=1}^{K} \left\| C_k - \frac{\text{Tr}(C_k)}{d} I_d \right\|_F^2$ and

$$Z = \begin{bmatrix} z_1, \ldots, z_B \end{bmatrix}^\top, \qquad \mu_k = \frac{1}{n_k} \sum_{i:y_i=k} z_i, \qquad C_k = \frac{1}{n_k - 1} \sum_{i:y_i=k} (z_i - \mu_k)(z_i - \mu_k)^\top$$

For the ImageNet dataset, given the significantly more complex underlying distribution, we add an additional regularizer, measuring the maximum mean discrepancy of each class conditional from a Gaussian distribution

$$\mathcal{L} = \mathcal{L}_{\text{MCR}^2} + \lambda_{\text{iso}} \cdot \mathcal{L}_{\text{iso}} + \lambda_{\text{MMD}} \cdot \sum_k \text{MMD}^2(z_k, \tilde{z}_k), \tag{96}$$

where $\tilde{z}_k \sim \mathcal{N}(\mu_k, I)$ and $\text{MMD}^2(z_k, \tilde{z}_k)$ is a kernel-based distance between two discrete distributions defined as follows

$$\text{MMD}^2(z_k, \tilde{z}_k) = \frac{1}{n^2} \sum_{i,j} k(z_i, z_j) + \frac{1}{m^2} \sum_{i,j} k(\tilde{z}_i, \tilde{z}_j) - \frac{2}{nm} \sum_{i,j} k(z_i, \tilde{z}_j) \tag{97}$$

where $k(\cdot, \cdot)$ is a positive-definite kernel function.

The choice of the kernel is the Gaussian Radial Basis Function (RBF) kernel

$$k(x, y) = \exp\left( -\frac{\|x - y\|^2}{2\sigma^2} \right)$$

During training, we freeze the FARE-4 backbone and we add, similarly to **?**, a pre-feature layer composed of Linear-BatchNorm-ReLU-Linear-ReLU. For feature head and cluster head, we utilize a Linear layer that maps the hidden to the feature dimension $d = 128$.

**Optimization, Initialization and Hyperparameters.** We initially warmup our pipeline by training the MCR$^2$ loss and then optimize simultaneously the feature cluster head using the MLC loss. Following Chu et al. (2023), we use the SGD optimizer for both the feature head and cluster head with learning rate equal to 0.0001, momentum set to 0.9 and weight decay set to 0.0001 and 0.005 respectively. All other hyperparameters remain the same to the ones used in Chu et al. (2023), thus referring the interested reader to the aforementioned related work.

13.2 ADDITIONAL EXPERIMENTS

**Separation of Classes.** We visualize the correlation of the latent embeddings of different classes showing the effectiveness of the $MCR^2$ loss in the CIFAR-10 dataset. As shown in Figure 4, such an encoder trained with the $MCR^2$ objective maps each class of input samples to points near a low-dimensional subspace, as the singular values of the mapped points drop quickly, while the mapped points from different subspaces tend to be orthogonal.

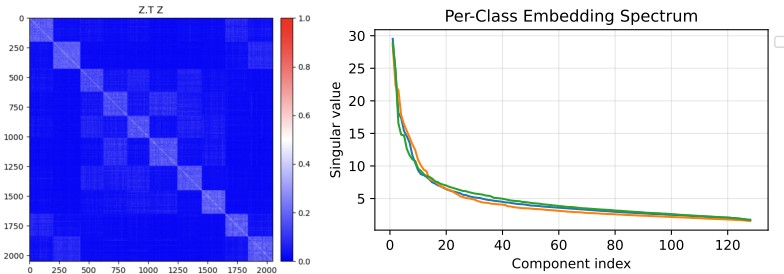

Figure 4: The correlation matrix and minimum eigenvalues of the latent space embeddings for the different classes for CIFAR-10.

**Empirical Validation of Sample Complexity.** In order to empirically validate the result of Theorem 3.2, we have first expressed the established sample complexity in terms of logarithms, as follows:

from $n = \tilde{\mathcal{O}}(\frac{d^{9/2}}{\epsilon^{9/2}})$, we know that there exists a constant $C > 0$ such that

$$
\begin{aligned}
n &\leq \frac{Cd^{9/2}}{\epsilon^{9/2}} \\
\Rightarrow \log \epsilon &\leq -\frac{2}{9}\log n + \log d + \frac{2}{9}\log C
\end{aligned}
\tag{98}
$$

We run multiple experiments for different sample sizes $n \in \{10, 100, 500, 1000\}$ and dimensions $d \in \{2, 10, 100\}$ and estimate in each experiment the distance of the learned certificate from the true certificate of robustness. We, next perform linear regression to estimate the coefficients $\alpha, \beta, \gamma$ in the following equality and compare with the ones of (98). We have found that and , thus validating empirically (1) and hence the sample complexity established in Theorem 4.2. CIFAR-10 We run multiple experiments for different sample sizes and dimensions and estimate in each experiment the distance of the learned certificate from the true certificate of robustness. We, lastly, performed linear regression to estimate the coefficients in the following equality and compared them with the ones in (1). We have found that

and , thus validating empirically (1) and hence the sample complexity established in Theorem 4.2. For the same example, we have plotted additionally the difference of the learned certified radius from the true one for different sample sizes and have shown how the certified radius scales with respect to the parameters and .

**On Gaussianity of the Latent Distribution.** To empirically validate that the used encoder maps the input distribution to a mixture of Gaussians, we apply Mardia's statistical normality test (Mardia, Biometrika, 57(3)), a well-known statistical test that evaluates whether a multivariate dataset departs from a Gaussian distribution. More specifically, Mardia's test computes two statistics:

1. multivariate skewness, which accounts for asymmetry
2. multivariate kurtosis, which evaluates whether the distribution's tail behavior matches the one of a Gaussian.

Under the null hypothesis, the data follow a multivariate normal distribution. The results show that the embeddings pass this test for all the classes, indicating that the class-conditional distributions are indeed conforming to Gaussians.

| Dataset | Mardia's Average Score | Percentage of Classes Passing Normality Test |
|---------|------------------------|----------------------------------------------|
| CIFAR-10 | 0.027 | 100% |
| ImageNet | 0.014 | 100% |

Table 4: Mardia's test results validate that the latent distribution of the encoder conforms with a mixture of Gaussian distributions.

**Synthetic Experiments.** We conduct experiments evaluating the robustness of the ELLIPS classifier in different Gaussian mixture settings. We test on Gaussian mixtures with different number of classes $K = \{2, 3, 5\}$, having different distance $R = \{2, 4, 5\}$ between the means of the classes and for isotropic and anisotropic covariance matrices.

We compare the certified accuracy of the proposed classifier with the method of Pal et al. (2023). We plot in Figure 5 the certified accuracy of both methods for the isotropic GMMs and in Figure 6 for anisotropic covariances. As shown in Figure 5 and Figure 6, our method outperforms the one in Pal et al. (2023) and closely approximates the empirical robust accuracy achieved by PGD attack.

Additionally, we compare the certified radius of Theorem 3.1 with the archetypal technique of randomized smoothing in different settings. As shown in Figure 7 and Figure 8, our method provides higher certified accuracy than randomized smoothing, indicating the tighter certification of the proposed radius of robustness.

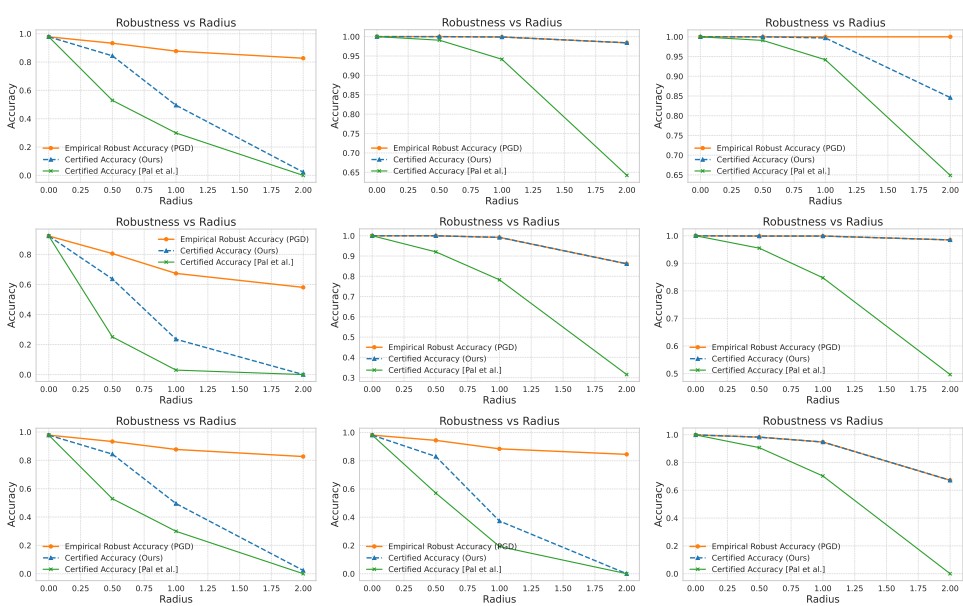

Figure 5: The proposed approach outperforms the method of Pal et al. (2023) in different Gaussian mixture settings. Each row corresponds to a GMM with isotropic covariances and different number of classes $K = \{2, 3, 5\}$, while each column to one with different separation distance $R = \{2, 4, 5\}$.

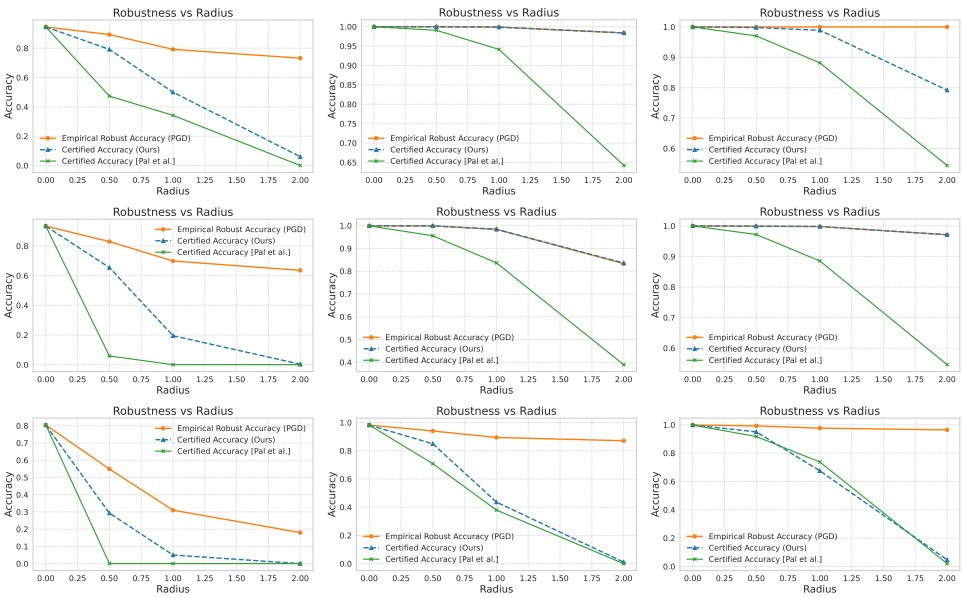

Figure 6: The proposed method outperforms the method of Pal et al. (2023) in different Gaussian mixtures with *anisotropic* covariance matrices. Each row corresponds to a GMM with different number of classes $K = \{2, 3, 5\}$, while each column to one with different separation distance $R = \{2, 4, 5\}$.

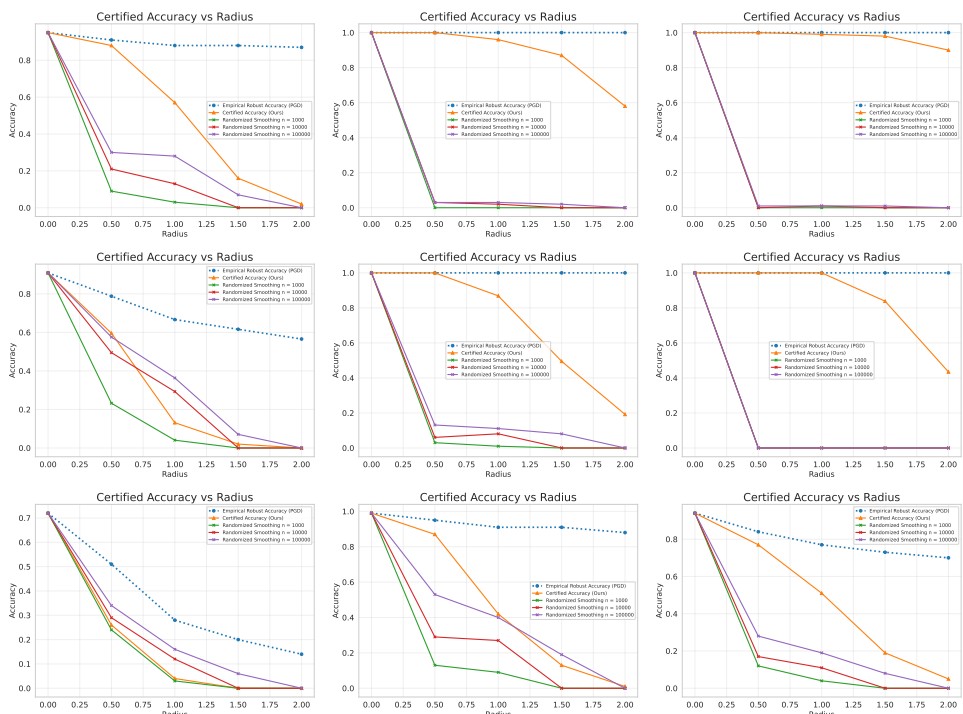

Figure 7: The proposed method achieves competitive robust accuracy in comparison to certified accuracy than randomized smoothing in different Gaussian mixture settings. Each row corresponds to a GMM with isotropic covariances and different number of classes $K = \{2, 3, 5\}$, while each column to one with different seperation distance $R = \{2, 4, 5\}$.

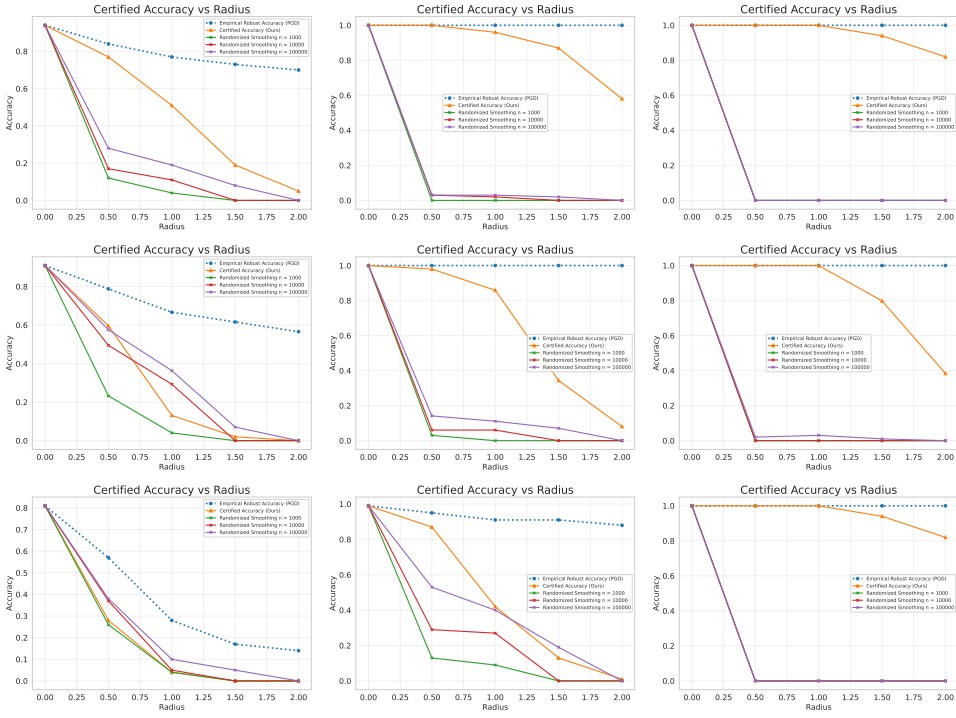

Figure 8: Comparison of our method with randomized smoothing for different mixture of Gaussians with *anisotropic* covariances. The proposed method performs competitively against randomized smoothing even when less number of Monte Carlo samples are used.

