$$
\begin{align}
p_i(S_i) &\geq 1 - \delta \tag{9}\\
\mathrm{Vol}(S_i) &\leq Ce^{-\epsilon} \tag{10}
\end{align}
$$

where $p_i$ is the density function of $\mathcal{D}_i$. We, first, define the set $S_i$ on which each Gaussian distribution $\mathcal{D}_i$ localizes. To do so, consider the probability density function

$$
p_i(x) = \frac{1}{\sqrt{(2\pi)^d \det(\Sigma_i)}} e^{-\frac{1}{2}(x-\mu_i)^T \Sigma_i^{-1}(x-\mu_i)}
$$

and the $c$ level-set

$$
A_c = \{x \in \mathcal{X} : p_i(x) \geq c\} \tag{11}
$$

for some fixed $0 < c < p_i(\mu_i)$. We want to select $c$ such that at least $1 - \delta$ of the mass is included in this level set, so that inequality (9) holds. Note that for a fixed $c$ the level-set is an ellipsoid, as it holds that

$$
\begin{align}
p_i(x) &= c \\
\iff \ln p_i(x) &= \ln c \\
\iff (x - \mu_i)^T \Sigma_i^{-1}(x - \mu_i) &= -[2\ln(c) + d\ln(2\pi) + \ln(\det(\Sigma_i))]
\end{align}
$$

Letting $r_i^2 = -[2\ln(c) + d\ln(2\pi) + \ln(\det(\Sigma_i))]$ for any $0 < c < p_i(\mu_i)$, the level set in (11) can be equivalently written as

$$
S_i = \left\{x \in \mathcal{X} : (x - \mu_i)^T \Sigma_i^{-1}(x - \mu_i) \leq r_i^2 \right\}
$$

which is the set of points with Mahalanobis distance $d_M(x, \mu_i) \leq r_i$.

In order for (9) to hold, we want to find the level set $c$ of $\mathcal{A}_c$ or equivalently the radius $r_i$ of the set $S_i$ such that at least $1 - \delta$ of the mass of the Gaussian distribution $\mathcal{N}(\mu_i, \Sigma_i)$ is included in $S_i$

$$\int_{S_i} p_i(x) dx \geq 1 - \delta \tag{12}$$

The integral in (12) is the probability that a sample $x \sim \mathcal{D}_i$ lies inside the set $S_i$ and thus we get equivalently that the following should hold

$$\mathbb{P}_{x \sim \mathcal{D}_i}(x \in S_i) = \int_{S_i} p_i(x) dx \geq 1 - \delta \tag{13}$$

By a change of variables $y = \Sigma_i^{-1/2}(x - \mu_i)$, we can transform the density $p_i(x)$ inside the integral to the density of the standard $\mathcal{N}(0, I)$ Gaussian $f(x) = \frac{1}{\sqrt{(2\pi)^d}} e^{-\frac{1}{2}\|y\|_2^2}$ and thus the set $S_i$ can be equivalently written as

$$\hat{S}_i = \left\{ x \in \mathcal{X} : \|y\|_2^2 \leq r_i^2 \right\}.$$

Hence, inequality (13) after the change of variables $y = \Sigma_i^{-1/2}(x - \mu_i)$ requires

$$\mathbb{P}_{x \sim \mathcal{D}_i}\left( x \in \mathcal{X} : \|y\|_2^2 \leq r_i^2 \right) \leq 1 - \delta. \tag{14}$$

Note, now, that since $y = \Sigma_i^{-1/2}(x - \mu_i)$ follows the standard Gaussian distribution $\mathcal{N}(0, I)$, the random variable $\|y\|_2^2$ follows the chi-squared distribution with $d$ degrees of freedom. Hence, the left hand-side of (14) is exactly the cumulative probability distribution of the $\chi_d^2$ distribution up to $r_i^2$. Thus, in order for (14) to hold, the $r_i^2$ should be the $(1 - \delta)$-quantile of $\chi_d^2$, i.e.

$$\begin{aligned} F_{\chi_d^2}(r_i^2) &\leq 1 - \delta \\ \delta &\leq 1 - F_{\chi_d^2}(r_i^2) \end{aligned}$$

where $F_{\chi_d^2}$ is the cumulative distribution function of the $\chi_d^2$.

In order for inequality (10) to hold, we have that

$$\begin{aligned} \mathrm{Vol}(S_i) &\leq Ce^{-\epsilon} \\ \iff \frac{\pi^{d/2} r_i^d}{\Gamma\left(\frac{d}{2} + 1\right)} \sqrt{\det(\Sigma_i)} &\leq Ce^{-\epsilon} \\ \iff \epsilon &\leq \ln\left( \frac{\Gamma\left(\frac{d}{2} + 1\right) C}{\pi^{d/2} r_i^d \sqrt{\det(\Sigma_i)}} \right) \end{aligned} \tag{15}$$

where $\Gamma(\cdot)$ is the Gamma function. $\qquad\square$

# 9 PROOF OF THEOREM 2.2

*Proof.* In order to show that $\mathcal{D}$ is $(C, \epsilon, \delta, \gamma)-$strongly localized, we need to show that for each class conditional $\mathcal{D}_i, \forall i \in [K]$, there is a set $S_i \subseteq \mathcal{X}$ such that the following hold

$$\begin{aligned} p_i(S_i) &\geq 1 - \delta \tag{16} \\ \mathrm{Vol}(S_i) &\leq Ce^{-\epsilon} \tag{17} \end{