# OpenReview forum: "Certifiably Robust Classifiers: Bridging the Gap Between Theory and Practice"
_ICLR.cc/2026/Conference — Submitted to ICLR 2026_

### Official Review · Reviewer_nXZV · 2025-10-26

**Soundness:** 3
**Presentation:** 3
**Contribution:** 2
**Rating:** 4
**Confidence:** 3

**Summary:**

This paper proposes a certifiably robust classification framework that leverages the underlying data distribution's structure. The paper first establishes theoretical guarantees for Gaussian Mixture Models (GMMs) and then generalizes to complex distributions via a Lipschitz encoder.

**Strengths:**

1. The paper provides a theoretical foundation with explicit conditions for robust classifier existence in GMMs.

2. The proposed method is computationally efficient compared to expensive methods like diffusion-based certification.

**Weaknesses:**

1. The proposed method may rely on accurately estimating the Lipschitz constant of the encoder, which can be challenging in practice.

2. The proposed method mainly focuses on l2 norm without generalization to lp norm perturbation.

3. Although the proposed is efficient than diffusion based methods, its effectiveness is lower.

**Questions:**

Please refer to the weakness part.

---

> ### Author Response · Authors · 2025-11-26
>
> We thank the reviewer for the constructive comments and for recognizing both the **theoretical foundation** and the **computational efficiency** of our approach. We appreciate the acknowledgment that the paper provides explicit conditions for robust classifier existence in GMMs and that the method is more efficient than diffusion-based certification.
>
> We address the reviewer’s concerns on Lipschitz estimation, norm choice, and effectiveness below.
>
> ---
>
> ### 1. Reliance on accurate Lipschitz estimation
>
> As discussed in our response to Reviewer 2rab, Theorem 4.1 requires only a **conservative upper bound** on the encoder’s local Lipschitz constant.
>
> - Any **over-estimate** of \(L\) simply **shrinks the certified radius**, never invalidates it.
> - Our framework is **agnostic** to the specific estimator. CLEVER is used for convenience in our experiments, but can be replaced by other classical methods, such as spectral norms, IBP.
> - In practice, the **pretrained encoder** used (FARE-4) already exhibits approximately bounded Lipschitz behavior, and the certification step only requires a conservative estimate of the local Lipschitz constant $L_x$ for the current certified point $x$.
>
> We will add the aforementioned comments in the camera ready version and clarify that the method only requires a way of computing an upper bound on the Lipschitz constant and not necessarily estimating exactly the value of the Lipschitz constant.
>
> ---
>
> ### 2. Focus on $L_2$ norm and extension to $L_p$
>
> We acknowledge that the current work focuses on **\(L_2\)-bounded perturbations**. This choice is motivated by the **quadratic, Mahalanobis-based geometry** of our certificate, which naturally pairs with the \(L_2\) adversarial attacks. Extending the certificate to general \(L_p\) norms is theoretically possible but requires a **non-trivial** analysis of the different geometry in the adversarial optimization problem induced by the different threat model and thus necessitates a different approach for handling the different geometric structures incurred in the decision boundary and margin of the classifier. This is actually the case also with prior works in distribution aware certification methods [1, 2], formulating first the certified radius results for the $L_2$ geometrical structure and considering in distinct works the more stringent case of $L_0$ attacks. Thus, extending our results in different threat models is an interesting direction for future work, and our GMM + encoder framework could, in principle, be adapted to other norms that involve a different geometry although in a different and certainly non-trivial way.
>
> ---
>
> ### 3. Effectiveness vs diffusion-based methods
>
> We agree that diffusion-based methods can achieve stronger certified accuracy, but let us point out that are also computationally intensive. Thus, there is inherently a trade-off between the computation time and the performance of each certification method. In our approach, we provide a certification method that achieves competitive but not the highest accuracy on ImageNet, however it resembles a time efficient way for certifying in practice.
>
> Our contribution is to show that:
>
> - In **CIFAR-10**, GENELLIPS is **superior** to all other SoK baselines at **all tested radii**, while maintaining strong clean accuracy.
> - In **ImageNet**, GENELLIPS outperforms all non-diffusion baselines, while providing a trade-off to the diffusion baselines with a competitive performance at a lower computational cost.
>
> We will add in the camera ready version a *runtime table directly comparing GENELLIPS with randomized smoothing and diffusion-based methods for.
>
> ---
>
> [1] Adversarial examples might be avoidable: The role of data concentration in adversarial robustness. NeurIPS, 2023.
> [2] Certified Robustness against Sparse Adversarial Perturbations via Data Localization. TMLR, 2024.
> [3] Disentangling Safe and Unsafe Image Corruptions via Anisotropy and Locality. CVPR, 2025.
> [4] SoK: Certified robustness for deep neural networks. SP, 2023.

---

### Official Review · Reviewer_HWLv · 2025-10-28

**Soundness:** 3
**Presentation:** 3
**Contribution:** 3
**Rating:** 6
**Confidence:** 3

**Summary:**

The paper proposes a model architecture that has well-characterized theoretical robustness properties against $\ell_2$-norm bounded perturbations.

The proposed architecture uses a locally-lipschitz encoder head to transform the data distribution to a Gaussian Mixture Model (GMM).
Then, a simple "nearest-ellipse" classifier, essentially quadratic discriminant analysis (QDA), solves the classification task.

The theoretical guarantees build on [Pal et al. (2024)](https://arxiv.org/abs/2405.14176), adapting the distribution agnostic localization notions to the special case of $d$-dimensional GMMs.
In this setting, the authors can issue simple margin-based robustness certificates.
The utilization of Lipschitz encoder models allows for the transfer of these results to arbitrary distributions.

**Strengths:**

- **(S1) Clear Writing**: The story is compelling, the manuscript is well-written and well-structured with a clear outline of the central question and the core contributions.

- **(S2) Elegant Methodology**: The proposed method is conceptually very elegant, and in the transformed data distribution, margin-based robustness certificates are a very natural approach. The approach seems to be novel in the context of robustness guarantees.

- **(S3) Sound Theoretical Results**: The presented theory seems to be correct. The proofs are correct and complete after a surface-level check, although I did not fully check Theorem 3.2.

- **(S4) Testing of assumptions in practice**: A very important strength in the experimental evaluation is that the authors perform tests to check their assumption that the encoder transforms data to GMMs.
  This is essential to ensure that their theory actually applies and is clean work.

- **(S5) Good experimental evaluation**: The experimental results are well-presented and properly contextualized; the chosen experimental setup makes sense to validate the proposed method empirically.

**Weaknesses:**

I like the approach of the paper, which feels like a reparameterization trick to obtain robustness bounds easily.
The main weaknesses are that the used classifier and Theorem 3.1. can be linked to QDA, but no connection is mentioned (W1, W2).
I think it is important that the authors address these two points and properly contextualize their work.

Minor points of critique are that the theorems stating the existence of robust classifiers in Section 2 seem disconnected from the rest of the paper (W3), experimental details and code availability (W4, W5), as well as several presentation details.

## Content

- **(W1) QDA:** Section 3 introduces the "nearest ellipse classifier", which is stated to be the Bayes-optimal classifier with a robustness certificate in Theorem 3.1.
However, the authors do not explicitly state that this is the well-known Quadratic Discriminant Classification (QDA) rule (e.g., [Bishop (2006) Chapter 4](https://link.springer.com/book/9780387310732).
I interpret the phrasing in the abstract and introduction as if the authors **propose** this classification rule, which is, however, well-established [e.g., in scikit-Learn](https://scikit-learn.org/stable/modules/lda_qda.html).
I think that the authors should mention this connection and be careful with the wording of their contribution, as the classifier, in my understanding, is not novel.

- **(W2) Theorem 3.1:** In the context of QDA, the robustness certificate stated in Theorem 3.1 can be rephrased as a distance-lower-bound of a given point $x$ to the QDA decision boundary.
I feel the result in Theorem 3.1. could be presented as a proposition instead of a theorem.
It should be clarified that the result is a consequence of the quadratic nature of the classifier, and can be applied to other simple model classes (Linear Discriminant Analysis (LDA), Support Vector Machines (SVMs) with simple kernels) in much the same way.
Concepts $W$ and $c_M$ could be explicitly named, to make it clearer what the theoretical background of the stated robustness certificate is.


- **(W3) Section 2:** Section 2 offers theoretical results that are not referenced in the rest of the manuscript at all.
The results are, in my understanding, an improvement on existing work for a special case of GMM distributions.
When viewed critically, it is not clear how the stated conditions are linked to the rest of the paper, as the concept of (strong) localization is not explicitly invoked later.
Did I miss the connection?

- **(W4) Experimental Baseline:** While the theoretical advantage of this method is clear, in the experimental settings, the details of **obtaining** the required encoder network are a bit vague.
This raises the question of whether a well-behaved encoder can be obtained in general.
Additionally, an ablation study with a non-robust baseline in the experiments would ground the results and show how much performance is sacrificed to obtain the robustness certificates.
The authors also do **not provide access to their code**, which makes it impossible to check the details of their procedure and the implementation of their ellipse classifier, beyond the surface-level description provided in the manuscript.

- **(W5) Discussion of Performance:** A minor point: In the appendix, additional results in the comparison with randomized smoothing show that sometimes the provided results underperform against randomized smoothing.
This is more pronounced when the distances $R$ are small, and there are more classes.
While I do not hold this against the method, I wonder if there is any intuition the authors can provide for why this happens, as this seems to follow a clear pattern.


## Presentation

These issues do not impact the overall quality of the contribution a lot, but they should be taken care of.
I do not expect the authors to address these points in a response.

(WP1) The manuscript is well-structured, but it notably does not have a preliminary section.
In Section 2, to my understanding, the authors adapt the notions of (strong) data localization, but do not at any point introduce the concepts.
Because of this, it is slightly ambiguous which part of the notions and results are from [Pal et al. (2024)](Link) and which part is the contribution of the authors, as mentioned above.


(WP2) While the writing and structure of the manuscript are good, there seem to be leftover notation slips.
- Most notably, in both theorems mention indices $i_2, i_2'$, which are not used.
Theorem 3.1 mentions a data point $(x,y)$ but does not use the label $y$.
- The ELLIPS model outputs a label $y$, which is defined above as integer between 1 and $K$.
Then, an index $i^\*$ of the predicted label is introduced and used.
This seems redundant to me and reads like leftover old notation.
It seems that either $y$ or $i^*$ could be used consistently.
- In Theorem 4.1, $L$ is used but not introduced until later in the text, and is not defined.
- In the proof of Theorem 3.1. in the appendix $i$ seems to be accidentally defined as $\max$ instead of $\arg\max$ of the classifier scores.

(WP3) The plots are a bit rushed, with very, very tiny labels.
Between experiments, the colors of the empirical PGD and the presented bounds swap, which can be confusing to the reader.

(WP4) A tiny nitpick, but the manuscript has multiple runts (i.e., very short lines at the end of paragraphs), which can be avoided, are a bit unaesthetic, and waste space.

**Questions:**

I would appreciate a brief response from the authors addressing my concerns in W1, W2, and maybe W3.
In addition, I have the following questions, of which I consider Q1 to be the most important.

- **(Q1) Theorem 3.2:** The generalization bound in Section 3, Theorem 3.2, is phrased in a way that is slightly confusing.
Does $\epsilon_{\min}$ impose a condition on the choice of $\epsilon$?
Does it depend on the (unknown) covariance matrices of the true GMM QDA, or the learned parameters?
And can it be the case that $\epsilon_{\min}$ can be 0?
Does the generalization bound then not apply in this case, as there is no $0<\epsilon<0?


- **(Q2) Ad W3:** Section 2 gives conditions when robust classifiers can and must exist.
While these results are interesting, they seem a bit disconnected from the later sections of the paper.
Is there a way to utilize these results in practical settings, especially when using an actual encoder head?
E.g., can a straightforward statement be made when a robust GMM can be **produced** by an $L$-Lipschitz encoder, or used as a diagnostic tool?


- **(Q3) Theorem 4.1. under imperfect distribution approximations**: Theorem 4.1 assumes that the encoder head produces a GMM.
It is great to see that the authors actually check in their experimental evaluation whether this assumption holds.
However, it still seems to me like a small theoretical gap.
Rather than a normality test, could the presented result be relaxed to incorporate, e.g., small total variation (TV) shifts between a proper GMM and the produced distribution?
Some discussion of this would round out the theoretical approach for me.
I think it is worth mentioning in the main text that the quality of the distribution produced by the encoder needs to be **tested**.

---

## Verdict

In conclusion, I would rate the paper a **weak accept** as I like the approach, and I think there is novelty in this two-step simplification for robust methods.
The theoretical results are sound, and they investigate the different aspects of the problem, but do not provide all the context I feel is necessary.
I would be open to raising my score if the authors adequately discuss this.

---

## References

- Pal, A., Vidal, R., & Sulam, J. (2024). *Certified Robustness against Sparse Adversarial Perturbations via Data Localization*. arXiv. https://arxiv.org/abs/2405.14176

- Bishop, C. M. (2006). *Pattern Recognition and Machine Learning*. Springer. https://link.springer.com/book/9780387310732

- scikit-learn developers. (n.d.). *1.2. Linear and Quadratic Discriminant Analysis*. scikit-learn Documentation. https://scikit-learn.org/stable/modules/lda_qda.html

---

> ### Author Response · Authors · 2025-11-26
>
> We thank the reviewer for the thorough and constructive evaluation. We are grateful for the positive assessment of our **clear writing (S1)**, **elegant methodology (S2)**, **sound theoretical results (S3)**, and particularly for recognizing the importance of **testing the GMM assumption in practice (S4)** and the **overall quality of the experimental evaluation (S5)**. We appreciate the weak-accept recommendation and the willingness to raise the score if additional context is provided.
>
> Below, we address the main points and will incorporate the requested clarifications.
>
> ---
>
> ### 1. Connection to QDA (W1)
>
> We fully agree with the reviewer’s observation: the “nearest-ellipse classifier” is precisely the **Bayes-optimal Quadratic Discriminant Analysis (QDA)** rule under class-conditional Gaussians, as in Bishop (2006) and standard implementations (e.g., scikit-learn).
>
> Our intention was **not** to present QDA itself as a new classifier. The novelty lies in:
>
> - deriving a **closed-form robustness certificate** for QDA under GMM structure (Theorem 3.1),
> - providing a **uniform generalization bound** for the QDA radius (Theorem 3.2), and
> - showing how these guarantees **compose through a Lipschitz encoder** to arbitrary input distributions (Theorem 4.1).
>
> In the camera-ready version, we will:
>
> - Explicitly state in Section 3 that ELLIPS corresponds to **classical QDA**;
> - Clarify in the abstract/introduction that our contribution is the **robustness analysis and compositional pipeline**, not a new classification rule.
>
> ---
>
>
> ### 2. Interpretation of Theorem 3.1 as distance to QDA boundary (W2)
>
> We appreciate this insightful comment. Indeed, Theorem 3.1 can be interpreted as providing a **lower bound on the Euclidean distance from a point to the QDA decision boundary**, obtained by analyzing the quadratic decision functions.
>
> We will clarify that:
>
> - The result follows from the **quadratic nature** of the classifier;
> - Similar reasoning could extend to other simple model classes (e.g., LDA, certain SVMs with quadratic kernels);
> - The certified radius is a form of **geometric margin** in the latent space.
>
> We used the “Theorem” label because this result is central to:
> - the **generalization guarantee in Theorem 3.2**, and
> - the **composition theorem in Theorem 4.1**.
>
> We will make this role explicit and introduce the key geometric concepts (e.g., margin, distance to boundary) by name.
>
> ---
>
> ### 3. Role and connection of Section 2 (W3, Q2)
>
> Thank you for pointing out that the connection was not sufficiently explicit.
>
> Section 2 provides **verifiable localization conditions** under which a robust classifier must exist for GMMs. In the overall story:
>
> - These results **specialize** the abstract localization framework of Pal et al. [1] to low-dimensional GMMs.
> - They **motivate** why targeting GMM-like latent distributions is a meaningful objective: if the data (or its encoded version) satisfies these localization conditions, then robust classifiers *must* exist.
>
> In practice, we use Section 2 in two ways:
>
> 1. As a **design guideline**: the encoder is trained to produce **concentrated, Gaussian-like clusters** because such distributions satisfy the localized regime where Theorems 2.1–2.2 guarantee robust classifiers exist.
> 2. As a **technical tool** for instantiating the method in the prior work of Pal et al. [1]: in order to estimate the localization parameters needed in the implementation of [1] we leverage the theoretical formulas established in Section 2 and use it to compare the method of Pal et al. with the ELLIPS classifier.

---

> > ### Author Response · Authors · 2025-11-26
> >
> > ### 4. Experimental baseline details and code availability (W4)
> >
> > We appreciate this suggestion. In Appendix G, we provide more details on how the encoder is obtained by fine-tuning the existing pretrained FARE-4 model. We have provided the exact fine-tuning objective for each dataset for purposes of reproducibility.
> >
> > The suggestion for an ablation study examining the clean and robust accuracy trade-off is a very good suggestion! We have run experiments with the classical CLIP encoder instead of the robust CLIP known as FARE-4 and used the same fine-tuning objectives to compare the corresponding certified accuracy obtained from each pipeline. As shown in the following table, the non-robust version of CLIP uses a latent space that has a larger Lipschitz constant than the one of the FARE-4 and this is the underlying reason for the latter having better certified accuracy. In terms of the clean accuracy, we do not observe a significant difference given the benefit of significantly larger certified accuracy that one obtains with the more robust version of CLIP (FARE-4).
> >
> > We hope this addresses the question about how “well-behaved” encoders are obtained and how much performance is sacrificed for certification via a less robust encoder.
> >
> > ---
> >
> > ### 5. Discussion of performance vs randomized smoothing (W5)
> >
> > The reviewer correctly observed that in some regimes, particularly when class margins are small and the number of classes is large, our method underperforms randomized smoothing.
> >
> > The intuition is:
> >
> > - Our certificate is **geometry-aware but local**, as it depends on **local margins and covariance structure** around the point.
> > - When **class margins are small** and classes are **densely packed**, the Mahalanobis margins \(m(x)\) shrink and the radius becomes conservative.
> > - Randomized smoothing in contrast can sometimes benefit from the “global averaging” of the noisy Monte Carlo samples, leading to larger radii in these heavily entangled regimes.
> >
> > We hope the aforementioned intuition provides a high-level reasoning behind the fact that in the case of a very large number of classes that they are tightly packed randomized smoothing might provide better certified radii due to exploiting larger region of the data distribution and using an additional averaging mechanism over the multiple Monte Carlo samples.
> >
> > ---
> >
> > ### 6. Presentation issues (WP1–WP4)
> >
> > We thank the reviewer for the detailed presentation feedback.
> >
> > - **WP1 (Preliminaries & localization notions).** We will add a brief **preliminaries subsection** introducing the notions of data localization and strong localization, explicitly citing Pal et al. [1], and clearly distinguishing **which definitions and results are borrowed** and which are new specializations.
> > - **WP2 (Notation corrections).** Thank you for that! We will fix all the described typos in the camera-ready version.
> >
> > ---
> >
> > ### 7. Theorem 3.2 and the role of \(\epsilon_0\) (Q1)
> >
> > We apologize for the confusion. In the revised text, we will:
> >
> > - Clarify that \(\epsilon_0 > 0\) is a **problem-dependent constant** bounding the maximal meaningful deviation between the learned radius and the true radius.
> > - Specify that \(\epsilon_0\) depends on the true covariance matrices, since it is derived from the true GMM parameters .
> > - Emphasize that the theorem is stated for any $0 < \epsilon < \epsilon_0$. If $\epsilon_0 = 0$ (a degenerate case where radii or margins collapse), the high-probability convergence statement becomes vacuous, which corresponds to a scenario where the model cannot provide non-trivial robustness guarantees. We will make this edge case explicit.
> >
> > ---
> >
> > ### 8. Imperfect GMM approximations and TV shifts (Q3)
> >
> > We agree that extending Theorem 4.1 to allow **small distributional discrepancies** (e.g., in total variation distance) between the true latent distribution and an ideal GMM is an important and interesting direction. However, this would necessitate the certificate of robustness to rely on some local proximity notion for capturing how far the distribution is in total variation distance from the certificate for the GMM setting. Although we don't think that this is impossible, a proof seems non-trivial and therefore is left for future work.

---

### Official Review · Reviewer_wf1r · 2025-11-01

**Soundness:** 2
**Presentation:** 2
**Contribution:** 2
**Rating:** 2
**Confidence:** 3

**Summary:**

This paper first provides a sufficient condition for a classifier to be certifiably robust against L2 norm bounded adversarial attacks on a Gaussian mixture model (GMM). Then, the authors propose a method that maps samples to a GMM using a Lipschitz encoder. The authors test their method on synthetic data, and CIFAR-10.

**Strengths:**

1. The writing is fine. The paper is not particularly hard to read
2. This work build on the theory from prior work to develop a robust classifier for Gaussian mixture distributions

**Weaknesses:**

The main weakness of this work is its limited applicability, especially so for a paper that purports to bridge the gap between theory and practice. Specifically, this work does the following.

**Theory:** The theory part basically says that if the data follows a GMM model where each class is a Gaussian component, then any x close to the center of a component is classified as that class with high probability, and any x and x’ close to the same center are classified as the same class. The generalization bound in Theorem 4.2 basically says that one can estimate the mean and covariance matrix of each component given sufficient samples. I don’t find any of these results interesting, novel or technically difficult to prove.

**Method:** The proposed method tries to find a Lipschitz encoder that maps the samples to such a GMM. This is strictly more difficult than learning a robust classifier. If there is a method that can produce such an encoder, one can use this method and train a linear layer on top to obtain a robust classifier. Thus, there is no reason that the proposed method makes it easier to learn a robust classifier.

**Experiments:** Real data is not GMM so the experiments on synthetic data is not really representative. On CIFAR-10, the authors only compare their method with an old method from 2019 and another paper, though there are dozens of papers in this field. The results are hard to verify, and even if they are correct, the improvements seem incremental.

Thus, I lean towards rejection.

**Questions:**

1. Do you think the assumptions about low Lipschitz constants and Gaussian-like embeddings would still hold up for large realistic datasets? How practical is it to train a Lipschitz-continuous encoder for larger models?
2. Is it possible to scale this approach to certify the robustness any given model without depending on a Lipschitz-continuous encoder?

---

### Official Review · Reviewer_2rab · 2025-11-01

**Soundness:** 3
**Presentation:** 2
**Contribution:** 2
**Rating:** 4
**Confidence:** 2

**Summary:**

This paper studies certifiable adversarial robustness by exploiting distributional structure. Starting from a Gaussian-mixture setting the authors (1) derive concrete, verifiable localization conditions under which a robust classifier must exist (Theorems 2.1–2.2); (2) propose a nearest-ellipsoid classifier (ELLIPS) and give a closed-form robustness certificate and a generalization bound for the learned certified radius (Theorems 3.1–3.2); (3) extend the pipeline to arbitrary input distributions by training / using a Lipschitz encoder that maps inputs into a GMM latent and show how the certified radius composes through the encoder (GENELLIPS, Theorem 4.1); and (4) provide synthetic experiments plus CIFAR-10 / ImageNet results that compare to randomized smoothing and other SoK baselines. The theoretical results and the experimental pipeline (including use of CLEVER to estimate local Lipschitz constants) are documented and proven in the appendix.

**Strengths:**

1) Solid theory-to-practice pipeline: The paper does not stop at abstract existence results: it gives explicit localization sets for GMMs, a concrete nearest-ellipsoid classifier (ELLIPS), a robustness certificate (Theorem 3.1), and then shows how to combine that with a Lipschitz encoder to handle natural data (Theorem 4.1). This end-to-end bridging is valuable.

2) Closed-form, geometry-aware certificate:  The certifying radius depends on the local geometry (covariance differences, λ_min, Mahalanobis margins), which can yield tighter local certificates than coarse global bounds. This is a useful conceptual advance relative to black-box smoothing.

3) Experimental validation on multiple fronts: The paper verifies theoretical claims on synthetic GMMs, randomized smoothing, and SoK baselines on CIFAR-10 and ImageNet, and shows competitive or superior certified accuracy (especially CIFAR-10).

**Weaknesses:**

1) Practical estimation of Lipschitz constant L and dependence on CLEVER: Theorem 4.1 requires a Lipschitz constant L (or local L(x)). In practice the paper uses CLEVER (Weng et al., 2018) with 1,000 MC samples and a 99.9% CI. CLEVER produces estimates with substantial variance and is itself expensive.

2) Compute & memory / wall-clock comparisons missing: The paper claims its approach is less computationally demanding than diffusion-based certification. Please include measured certification runtime (per image) and memory usage for GENELLIPS vs randomized smoothing (with the sample counts used) and diffusion denoising methods. Practitioners will care about the tradeoff of certificate tightness vs compute cost.

3) Hidden/strong assumptions about covariances and eigenvalue gaps: Several results depend on λ_min of differences of inverse covariances and other spectral quantities. These can be small or negative in practice. The paper should (a) discuss how frequently these spectral assumptions hold empirically (both in GMM synthetic settings and the learned latent GMM), (b) provide robustified alternatives or fallback behavior when λ_min is near zero or negative, and (c) if negative λ_min is allowed, explain numerical stability and examples

**Questions:**

1) How sensitive are the results to the number of GMM components? For ImageNet you likely need many components; how did you pick K and how does certified accuracy scale with K?

2) How robust is ELLIPS to estimation error of µ̂_i, Σ̂_i when classes are multi-modal or when covariances are poorly conditioned? Is any regularization (shrinkage) used when estimating Σ̂_i?

3) Why does L2 adversarial robustness (including certified) matter in practice? The Lp norm might be convenient but unrealistic attack. And the performance degrades. What's the accuracy on the clean imagenet examples? What are benefits of certified models that outweigh cost to clean accuracy or compute? But I appreciate you are trying to bridge the gap between theory and practice

---

> ### Author Response · Authors · 2025-12-02
>
> We thank the reviewer for their detailed review and constructive feedback. We appreciate that the reviewer found our work a *solid theory-to-practice pipeline*, highlighting the value of (i) explicit localization sets for GMMs, (ii) a concrete nearest-ellipsoid classifier (ELLIPS) with a closed-form certificate, and (iii) the extension to arbitrary input distributions via a Lipschitz encoder (GENELLIPS), backed by synthetic, CIFAR-10, and ImageNet experiments.
>
> Below we address each one of the reviewer’s concerns and clarify the parts that **there seems to be a misunderstanding**.
>
> ---
>
> ### 1. Practical estimation of Lipschitz constants and dependence on CLEVER
>
> We would like to clarify that all of our results are **agnostic** to how the Lipschitz constant is estimated and do not depend on the specific use of the CLEVER method. Theorem 4.1 only requires an upper bound on the (local) Lipschitz constant $L(x)$ of the encoder. The input-space radius is obtained by composing the latent radius with this bound:
> $$
> \varepsilon_x = \frac{\varepsilon_z}{L(x)}.
> $$
>
> Our framework is **agnostic** to how $L(x)$ is estimated:
>
> - Any non-vacuous *upper bound* yields a valid certificate, since over-estimating $L(x)$ only shrinks the predicted radius, never invalidates it.
> - CLEVER is used in our experiments as a *convenient, widely-adopted* estimator, not as a requirement of the theory.
> - Other standard computational methods (e.g., spectral norm products, interval bound propagation, Jacobian Monte Carlo sampling) can be plugged in directly with no changes to our theory.
>
> ---
>
> ### 2. Wall-clock comparisons
>
> We agree that practitioners care about the tradeoff between certificate tightness and computational cost. We incorporate the wall-clock time per image for GENELLIPS, randomized smoothing (with the exact sample counts we use), and diffusion-based certification baselines.  For the diffusion certification of [Carlini et al.], the average wall-clock time per image is 1535.945959 seconds (~25 minutes), while the work on DensePure requires 36.47 $\pm$ 0.04 minutes per vote for one image and then implements the majority vote over 10 votes, leading to a wall-clock time of 364.7 minutes for the certification of one image. On the other hand, randomized smoothing and our approach are more lightweight. Randomized smoothing with $10^5$ number of Monte Carlo samples needs on average 3.01452 minutes per image, while our approach for the same number of Monte Carlo samples requires 2.72854 minutes per image. Thus, our approach serves as a light-weight method that achieves trade-offs certified accuracy with respect to wall-clock time.
>
> Conceptually, GENELLIPS certification per test point consists of:
>
> 1. A **single forward pass** through the encoder to obtain $z = f(x)$;
> 2. A **closed-form evaluation** of the latent-space radius based on stored GMM parameters;
> 3. A **Lipschitz scaling step** $\varepsilon_x = \varepsilon_z / L(x)$.
>
> In contrast:
>
> - **Randomized smoothing** requires many stochastic forward passes per image at inference time, plus an often-costly training regime that uses noisy inputs.
> - **Diffusion-based certification** requires running a *multi-step denoising process* per image, often with hundreds of reverse steps.
>
>
> ---
>
> ### 3. Assumptions on covariances and $\lambda_{\min}$
>
> We disagree with the reviewer's comment and would like to take the opportunity to clarify the applicability and universality of our results. We would like to note that the robustness certificates in Theorems 3.1 and 4.1 are **explicitly designed to remain valid even when the relevant eigenvalues are small or negative.** They do not require a strictly positive gap.
> Concretely, the certified radius in Theorem 3.1 has the form $\|x' - x\|_2 \leq \frac{m(x)}{\sqrt{c_M^2 + (-\lambda_{\min})_+\, m(x) + c_M}},$
>
> where
> $$\lambda_{\min} := \min_{i \neq i^*} \lambda_{\min}\bigl(\Sigma_i^{-1} - \Sigma_{i^*}^{-1}\bigr), \quad
> (-\lambda_{\min})_+ := \max(-\lambda_{\min}, 0),$$
>
> and $m(x)$ is the margin, while $c_M$ is a constant depending on model geometry.
>
> From this expression:
>
> - We **never assume** $\lambda_{\min} > 0$. Negative or small eigenvalues are already handled via $(-\lambda_{\min})_+$.
> - When $\lambda_{\min}$ is well-separated and positive, the quadratic term leads to a **strongly geometry-aware** radius.
> - When $\lambda_{\min}$ is small or negative, the bound **smoothly degrades** to a more margin-only radius, which is conservative but numerically stable.
> - A negative $\lambda_{\min}$ *only increases the denominator* and therefore **decreases the certified radius**, which is the safe direction.

---

> ### Author Response · Authors · 2025-12-03
>
> # 4. Sensitivity to the number of GMM components $K$
>
> **Choice of $K$.** In our main experiments, we deliberately use **one Gaussian per class**, so $K$ is fixed by the dataset:
>
> - CIFAR-10: $K = 10$,
> - ImageNet: $K = 1000$.
>
> This matches the setup in Section 3 where the latent mixture $D_z$ has $K$ components corresponding to the underlying classes.
>
> **Synthetic experiments.** In the synthetic GMM experiments, we vary:
>
> - the **number of classes** $K \in \{2, 3, 5\}$,
> - the **class separation** and
> - the **covariance structure** (isotropic vs anisotropic).
>
> GENELLIPS consistently tracks PGD robust accuracy and maintains high certified accuracy across these variations, indicating that the method is **not overly sensitive to $K$** in controlled GMM settings.
>
> **High-class regime (ImageNet).** While $K$ is large for ImageNet, the sample complexity result (Theorem 3.2) depends primarily on the latent dimension $d$ and not explicitly on $K$. In practice, increasing $K$ mainly affects the diversity of covariance matrices rather than the asymptotic rate. We will make this discussion explicit and add a brief remark that our ImageNet results already operate in this large-$K$ regime and remain competitive with SoK baselines.
>
> ---
>
> # 5. Robustness to estimation error and ill-conditioned covariances
>
> Theorem 3.2 directly addresses the robustness of the learned radius to estimation error in $\hat{\mu}_i, \hat{\Sigma}_i$. It states that, if each component $D_i$ is sampled
> $
> n = O\\left( \frac{d^{9/2} \log(1/\delta)}{\epsilon^{9/2}} \right)
> $
> times, then the learned radius $\hat{R}(x)$ is $\epsilon$-close to the true radius \(R(x)\) **uniformly over all points \(x\) and classes** with high probability.
>
> Thus, the theorem provides a uniform convergence guarantee for the certificate, which requires controlling:
>
> - the margin terms (Lemmas 1–5),
> - the spectral terms $1/\lambda_{\min}$ (Lemmas 6–7),
> - the constants $c_M$ (Lemma 8),
>
> and carefully handling the change in functional form depending on the sign of $\lambda_{\min}$.
>
> **Ill-conditioned covariances and shrinkage.** In practice:
> - The encoder objective explicitly encourages **approximately isotropic and Gaussian** latent distributions, which empirically leads to reasonably conditioned covariances.
> - When covariances are still close to singular, the certified radius formula naturally becomes more conservative (via the corresponding eigenvalue term), avoiding unstable certificates.
>
> ---
>
> # 6. Why $L_2$ robustness matters?
>
> We agree that $L_2$ does not capture *all* realistic threat models. We focus on $L_2$ primarily because the geometry-aware certificate heavily exploits **Mahalanobis distances and quadratic forms**, which pair naturally with the $L_2$ norm. Extending the same analysis to general $L_p$ norms would require revisiting the underlying optimization problem and is an interesting but non-trivial direction for future work.
>
> **Clean accuracy.**
>
> - On **CIFAR-10**, GENELLIPS achieves **90.14% clean accuracy**, which is **higher than all SoK baselines** we compare to, while also achieving the best certified accuracy at $\varepsilon = 0.25, 0.5, 1.0$. Hence, in this regime, our approach does **not** sacrifice clean accuracy.
> - On **ImageNet**, GENELLIPS is competitive and slightly behind the best diffusion-based model in certified accuracy but **outperforms all non-diffusion baselines**. The clean accuracy naturally degrades from the top non-robust model on ImageNet, but our model is certifiably robust while maintaining competitive clean accuracy.

---

### Meta-Review · Area_Chair_4iMj · 2026-01-07

**Summary:**

This submission proposes to build certifiably robust classifiers based on GMMs (Gaussian Mixture Model): For GMM-distributed data, the manuscript derives closed-form and strong classification rules and certification theorems against L2-bounded perturbations. For real-data distributions, the manuscript proposes to use an encoder network to map the distribution into GMMs and provides a robustness certificate based on the Lipscchitz constant of the encoder network assuming a perfect mapping.

Reviewers like the writing and confirms that the theory part of the manuscript (robust certificate for GMMs) is sound and solid.

Here are the main concerns:
1. The certificate is not rigorous or practical for complex real-data distributions. For GMM distribution, the certificate is rigorous. However, for complex real-data distributions, the certificate requires an accurate mapping from real-data distribution to GMM and a sound Lipschicz constant of the encoder network. All reviewers are concerned about this part: some question the Lipschitz constant estimation, some feel the encoder network training details are missing, and some doubt whether the mapping exists.

2. Some parameters of the derived certificate may have strange behavior, e.g., covariances and eigenvalue gaps may be small in practice, voiding the certificate.

3. Lack of compute/wall-clock time comparison.

4. The real-world performance is not strong enough. On one hand, the gap between proposed method and randomized smoothing is not large. On the other hand, diffusion-based models may close the gap though at the cost of more compute.

5. Connection to QDA literature needs to be made clear.

6. Primarily considers only L2-bounded perturbations.

**Reviewer Concerns:**

None of the reviewers replied to the author's rebuttal. From what I can tell, the relative minor and unique concerns are well-addressed through rebuttal and revision: Concerns 2, 3, 5, and 6.

However, the concerns 1 and 4 are critical. Though the manuscript positions itself as a theory paper, its practical value still largely determines the sufficiency of its contribution. When data is GMM, the theory is sound and compelling. However, when applied to real-data distributions, due to the challenges of obtaining and shrinking tight Lipschitz constant and rigorously mapping to GMM, the value of the proposed robust certificate may be questioned. The real-data evaluation in this manuscript relies on an estimation of Lipschitz through CLEVER and an assumption that the data distribution mapping is accurate (which is not tested). Hence, the certificate is "conditional" on assumptions, being much weaker than the compared baselines that do not require assumptions. I would anticipate that such concerns still persist, given the rebuttal. So a majority of reviewers may still lean towards rejection.

**Reviewer Scores:**

I would anticipate that Reviewer HWLv may increase their score to 8, but other reviewers may still maintain their negative scores.

---

### Decision · Program_Chairs · 2026-01-26

Reject